# Dispersed surface Ru ensembles on MgO(111) for catalytic ammonia decomposition

Huihuang Fang[1,8], Simson Wu[1,8], Tugce Ayvali[1], Jianwei Zheng[1], Joshua Fellowes[1], Ping-Luen Ho [1], Kwan Chee Leung[1], Alexander Large [2], Georg Held[2], Ryuichi Kato[3], Kazu Suenaga[3], Yves Ira A. Reyes[4], Ho Viet Thang[5], Hsin-Yi Tiffany Chen [4,6,7] & Shik Chi Edman Tsang [1] ✉

Ammonia is regarded as an energy vector for hydrogen storage, transport and utilization, which links to usage of renewable energies. However, efficient catalysts for ammonia decomposition and their underlying mechanism yet remain obscure. Here we report that atomically-dispersed Ru atoms on MgO support on its polar (111) facets {denoted as MgO(111)} show the highest rate of ammonia decomposition, as far as we are aware, than all catalysts reported in literature due to the strong metal-support interaction and efficient surface coupling reaction. We have carefully investigated the loading effect of Ru from atomic form to cluster/nanoparticle on MgO(111). Progressive increase of surface Ru concentration, correlated with increase in specific activity per metal site, clearly indicates synergistic metal sites in close proximity, akin to those bimetallic $N_2$ complexes in solution are required for the stepwise dehydrogenation of ammonia to $N_2/H_2$, as also supported by DFT modelling. Whereas, beyond surface doping, the specific activity drops substantially upon the formation of Ru cluster/nanoparticle, which challenges the classical view of allegorically higher activity of coordinated Ru atoms in cluster form ($B_5$ sites) than isolated sites.

The ammonia decomposition has been subjected to a new opportunity for the potential use of $NH_3$ as a hydrogen storage medium[1–3]. Ammonia can be easily liquefied at a pressure of 8 bar at 20 °C, leading to a high energy density (4.25 kWh/L)[4,5]. It circumvents the safety problem in long-distance transport of $H_2$ and is readily available as C-free energy carrier to be used in stationary and mobile applications upon decomposition via wind and solar power[6]. However, catalytic sites and mechanism are yet far from clear. Many researchers commonly adopt fundamental steps from the reversed reaction: the ammonia synthesis from respective nitrogen and hydrogen despite the

fact that the reaction conditions can be significantly different from each other. Many transition metals such as Ni can give good catalytic activity for $NH_3$ decomposition at relatively evaluated temperatures and has been the focused for recent investigation due to low cost in large scale operation[7–9]. Ruthenium (Ru) has been well-studied as an efficient metal for ammonia decomposition at low temperatures due to its catalytic performance with many folds higher activity than other transition metals. Ilaria systematically summarized the recent progress in developing Ru based catalysts for ammonia decomposition and these Ru catalysts demonstrated a variety of catalytic performance due

[1]The Wolfson Catalysis Centre, Department of Chemistry, University of Oxford, Oxford OX1 3QR, UK. [2]Diamond Light Source, Didcot OX11 0DE, UK. [3]National Institute of Advanced Industrial Science and Technology (AIST), Central 5, 1-1-1 Higashi, Tsukuba 305-8565, Japan. [4]Department of Engineering and System Science, National Tsing Hua University, Hsinchu 300044, Taiwan. [5]The University of Danang, University of Science and Technology, DaNang 550000, Vietnam. [6]College of Semiconductor Research, National Tsing Hua University, 101, Sec. 2, Kuang-Fu Road, Hsinchu 300044, Taiwan. [7]Department of Material Science and Engineering, National Tsing Hua University, Hsinchu 300044, Taiwan. [8]These authors contributed equally: Huihuang Fang, Simson Wu. ✉e-mail: edman.tsang@chem.ox.ac.uk

to their different supports, metal structure and particle size[9]. It is found that Ru is also very structure-sensitive for the ammonia decomposition[10–12]. The sensitivity is thought to be stemmed from the chemistry occurring primarily on Ru cluster: the B$_5$-type sites, composed of three Ru atoms in the surface layer and two additional Ru atoms in a sub-surface layer, which ensures the activation of dinitrogen in stereospecific manner to atomic nitrogen species in line with the molecular orbitals (dissociative pathway). Previous studies demonstrated that the number of B$_5$ sites is highly dependent on particle size and shape, reaching at the maximum values at 1.8–3 nm and ~7 nm for hemispherical nanoparticles (NPs) and elongated NPs, respectively[13–15]. Much effort has been devoted to enrich the number of B$_5$ sites via particle control; nevertheless, the formation of Ru NPs lessens the atomic economy and limits the large-scale application due to the high cost of Ru[16–19]. In addition, traditional Ru-based catalysts are easily prone to hydrogen poisoning (they are also active to activate dihydrogen) in both forward and backward reactions under mild reaction conditions[20–22]. The typical hydrogen reaction orders for ammonia synthesis and decomposition over Ru catalysts are negative, indicating the blockage of Ru active sites by hydrogen adatoms and eventually preventing the chemisorption of nitrogen species[23–26]. Therefore, developing a reliable Ru catalyst and support such as carbon nanotubes to reduce strong hydrogen adsorption and maximize the efficiency in ammonia chemistry is of great importance[21,27]. On the other hand, reversible activation of N$_2$/H$_2$ to and from NH$_3$ is a longstanding and significant research focus within homogeneous catalysis community, which depicts a very different discrete active sites from the Ru cluster B$_5$ site in solid state. Ligand stabilized bi or multinuclear redox site systems to coordinate N$_2$ via diazenido-, hydrazido-, or nitrido-metal species are identified[28]. It would be important to compare the activity of surface dispersed 2 or more Ru sites akin to homogenous bi or multinuclear sites with B$_5$ sites and their hydrogen removal ability at the interface of support surface. Such approach may provide paramount hints to achieve higher activity and higher atom economy regarding the use of Ru atoms in catalysis.

Herein, we report the detailed loading effect of an atomically-dispersed Ru atoms on polar MgO (111) facet support (a high energy facet with alternative stacking of atomic layers of O$^{2-}$ and Mg$^{2+}$) that can efficiently catalyze the decomposition of ammonia in highly hierological manner. It is presented that the atomically dispersed surface Ru ensembles on MgO(111) surface gives an optimal turnover frequency (TOF) value of 3.43 s$^{-1}$ for ammonia decomposition at 400 °C, higher than those of allegorically more active Ru cluster/nanoparticle counterparts on MgO(111) (2.1–2.9 s$^{-1}$) in classic view and the ultralow-loading Ru/MgO(111) with isolated Ru atoms (1.67 s$^{-1}$). A series of characterizations including Scanning Transmission Electron Microscopy (STEM), X-ray diffraction (XRD), X-ray absorption fine structure spectroscopy (EXAFS), X-ray Photoelectron Spectroscopy (XPS), temperature-programmed desorption (TPD), temperature-programmed surface reaction (TPSR), In-situ diffuse reflectance infrared Fourier transform spectroscopy (DRIFTS) and Ambient Pressure XPS (AP-XPS) and DFT calculations were employed to investigate the interactions between Ru and polar MgO(111) facet and with neighboring Ru sites. Results clearly demonstrate the synergetic roles of individual Ru atoms in close proximity for ammonia activation and hydrogen migration to MgO (111) facets.

## Results

### Material synthesis and characterization

MgO is long empirically found to be one of the most active supports to host transition metals as catalysts in ammonia synthesis and decomposition[29–32]. It has also been shown that polar metal oxide surfaces including MgO could give unique catalytic properties due to their high surface energy and spontaneous electrostatic polarization along specific axes[33–35]. Very recently, atomic dispersion of transition metal atoms on MgO facet supports due to the strong electrostatic affinity for metal ions and surface oxygen anions has been successfully prepared[36–38]. The chemistry of single transition metal atoms whether they are stabilized by ligands or support surface sites as solid ligand groups have been compared and discussed[39–41].

Following these studies, it would also be exciting to carefully investigate the general support effect for Ru in varying loading on surface of different MgO facets in terms of specific activity in catalytic ammonia decomposition. In this regard, MgO nanosheets with preferential exposed (111), (110) and (100) facets were synthesized. MgO(111) faceted sample was prepared by integrated processes of surfactant-assistant hydrothermal treatment and controlled calcination as compared to MgO(110) and commercial MgO, see experimental for details. The catalytic performances of MgO supported Ru catalysts (Ru/MgO) in NH$_3$ decomposition are presented in Table S1. The superior performance of the Ru/MgO(111) catalyst is clearly demonstrated by the comprehensive comparison to the literature results under similar reaction conditions, as summarized in Table S1. The specific activities of Ru on different supports in term of mmol H$_2$ formed per g$_{Ru}$ per min and TOF values are compared under defined ammonia flow rate, metal loading and temperature. As seen, there is a clear support effect: supports that are known to facilitate the removal of hydrogen from overlying Ru (hydrogen spillover), such as C12A7:e$^-$, BHA and CNTs, give higher activity than that of inert supports (SiO$_2$, Al$_2$O$_3$)[42]. As stated, the MgO is empirically found to give high activity in both ammonia decomposition and synthesis; however, the simple electrostatic attraction and repulsion of Mg$^{2+}$ and O$^{2-}$ ions in NaCl structure with corresponding Madelung charge distribution without anticipated electronic interaction to give good support effect is not yet clear. Here, we have found that MgO(111) facet as support for Ru displays a superior activity: for 3.1wt.% Ru, under WHSV of 30,000 mL g$_{cat}^{-1}$ h$^{-1}$ at 425 °C, an equilibrium conversion is recorded and at 450 °C with higher WHSV of 60,000 mL g$_{cat}^{-1}$ h$^{-1}$ the highest specific activity of 1777.4 mmol H$_2$ g$_{Ru}^{-1}$ min$^{-1}$ with a TOF value of 4.91 is achieved as compared to all other support materials.

The formation of MgO hexagonal nanosheets with a large amount of exposed (111) facets was imaged by TEM and depicted in Fig. S1a[43]. X-ray diffraction analysis confirmed the typical rock-salt structure of MgO (Fig. 1a). To identify the presence of (111) facets, controlled assemblies of anisotropic MgO nanosheets were employed by using polar ethanol solvent and non-polar hexane solvent as dispersants. It can be seen that the ethanol-dispersed MgO(111)-PS exhibits two stronger peaks indexed as (111) and (222) diffractions with lower (200) and (220) reflections as demonstrated in Fig. 1a. On the contrary, the hexane-dispersed MgO(111)-NPS displays a reversed trend, which reveals the preferential orientations in MgO(111) samples[44]. Considering the geometry of MgO(111) nanosheet, the top and bottom faces are the (111) planes while the side faces are (220) planes based on the Bragg's law and scattered beam collection, indicating the successful synthesis of MgO with maximal exposed (111) facets (Fig. 1b). For comparison, the MgO(110) and MgO(100) with preferential exposure of (110) and (100) facets were also prepared based on previous reports[45,46], as evident in Fig. S1b, c.

The Ru supported MgO catalysts were synthesized via decomposition of Ru$_3$(CO)$_{12}$ onto MgO supports to avoid the deconstruction of MgO surface during Ru immobilization[47,48]. As shown in Fig. 1c, all as-prepared Ru catalysts display similar diffraction patterns to those of parent MgO support with no obvious diffractions ascribed to Ru species. This finding reveals that the deposition of Ru has a negligible effect on MgO structure and the Ru species are evenly dispersed on these supports. No Ru NPs was found in Ru/MgO(111) from the typical TEM image shown in Fig. S2. As evident in Fig. 1d, Ru species remain highly dispersed with

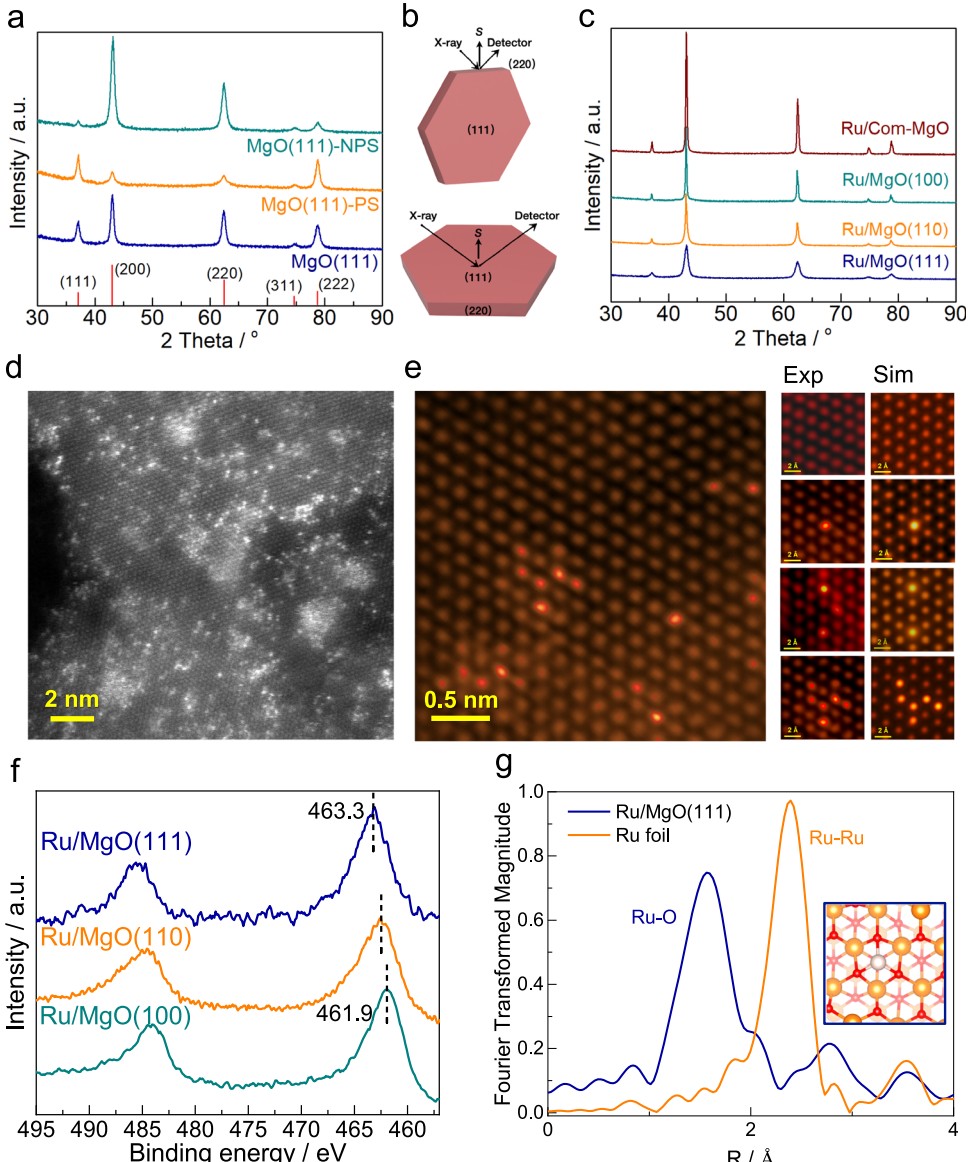

**Fig. 1 | Characterizations of Ru/MgO(111), Ru/MgO(110) and Ru/MgO(100).**
**a** XRD patterns of MgO(111) samples prepared in different solvents. Note: MgO(111)-PS and MgO(111)-NPS are assembled in ethanol and n-hexane as the polar solvent (PS) and non-polar solvent (NPS), respectively; **b** illustration of XRD patterns from ideally aligned MgO(111) nanosheets perpendicular and parallel to the diffraction vector s; **c** XRD patterns of 3.1 wt.% Ru supported on MgO with different predominant exposed facets; **d** HAADF-STEM images of 3.1 wt.% Ru/MgO(111), the scale bar in yellow is 2 nm; **e** HAADF-STEM images with observed single Ru atoms in the form of single Ru atoms, 2×Ru atoms, 3×Ru atoms and other close-by surface Ru atoms; the scale bar in yellow is 0.5 nm; Exp: experimental image, Sim: image simulation (see SI); **f** Ru 3p XPS profiles of samples, the scale bar in yellow is 2 Å; **g** Fourier transform of $k^3$-weighted Ru K-edge of EXAFS spectra of the Ru/MgO(111). The Ru metal foil is also included here for reference; The insert figure is simulated model of Ru coordinating with 3-oxygen atoms (trigonal site) on (111) oxygen terminated surface.

plenty of single "atoms" scattered on surface and as a 2D raft-like structure at higher coverage (no 3D aggregation and no metallic lattice is observed) on MgO(111) support. Further STEM image (Fig. 1e) demonstrates that Ru was atomically-dispersed in the form of single atoms, 2×Ru atoms, 3×Ru atoms and other close by surface Ru atoms with 2D islands of atomic thickness in Ru/MgO(111). The Fast-Fourier-Transform (FFT) pattern shown in Fig. S3 indicates that the Ru atom is located on the exposed MgO(111) facet where a typical Ru−Ru distance is measured to be around 5.1 Å, much longer than a typical Ru−Ru bond of 2.7 Å in metallic Ru. In contrast, Ru NPs are clearly seen in Ru/MgO(110) and Ru/MgO(100) samples (Fig. S4) along with atomically-dispersed Ru atoms. To investigate the electronic properties of Ru species, the Ru 3p profiles were obtained by XPS measurement, as shown in

Fig. 1f. The Ru displays typical 3p$_{3/2}$ peaks at 463.3 eV in Ru/MgO(111) sample, which is corresponding to Ru$^{2+}$ species. On the other hand, Ru/MgO(110) and Ru/MgO(100) display peaks with lower binding energies at 462.1 eV and 461.9 eV, respectively, corresponding to metallic Ru$^0$ species. The unique positive charge of Ru species in Ru/MgO(111) is due to the adaptation of the most stable configuration of Ru$^{2+}$ with terminal O$^{2-}$ (charge transfer) in MgO(111) surface with strong affinity, resulting in atomically dispersion of Ru in MgO(111). For identification of precise position of Ru atoms on MgO(111), high angle annular dark field (HAADF) imaging by 60 kV STEM was carried out. Fig. S5a–c shows the HAADF-STEM images of Ru/MgO(111) observed from [111], [110] and [100] directions. Brighter contrast dots compared with surrounding top O sites are ascribed to the

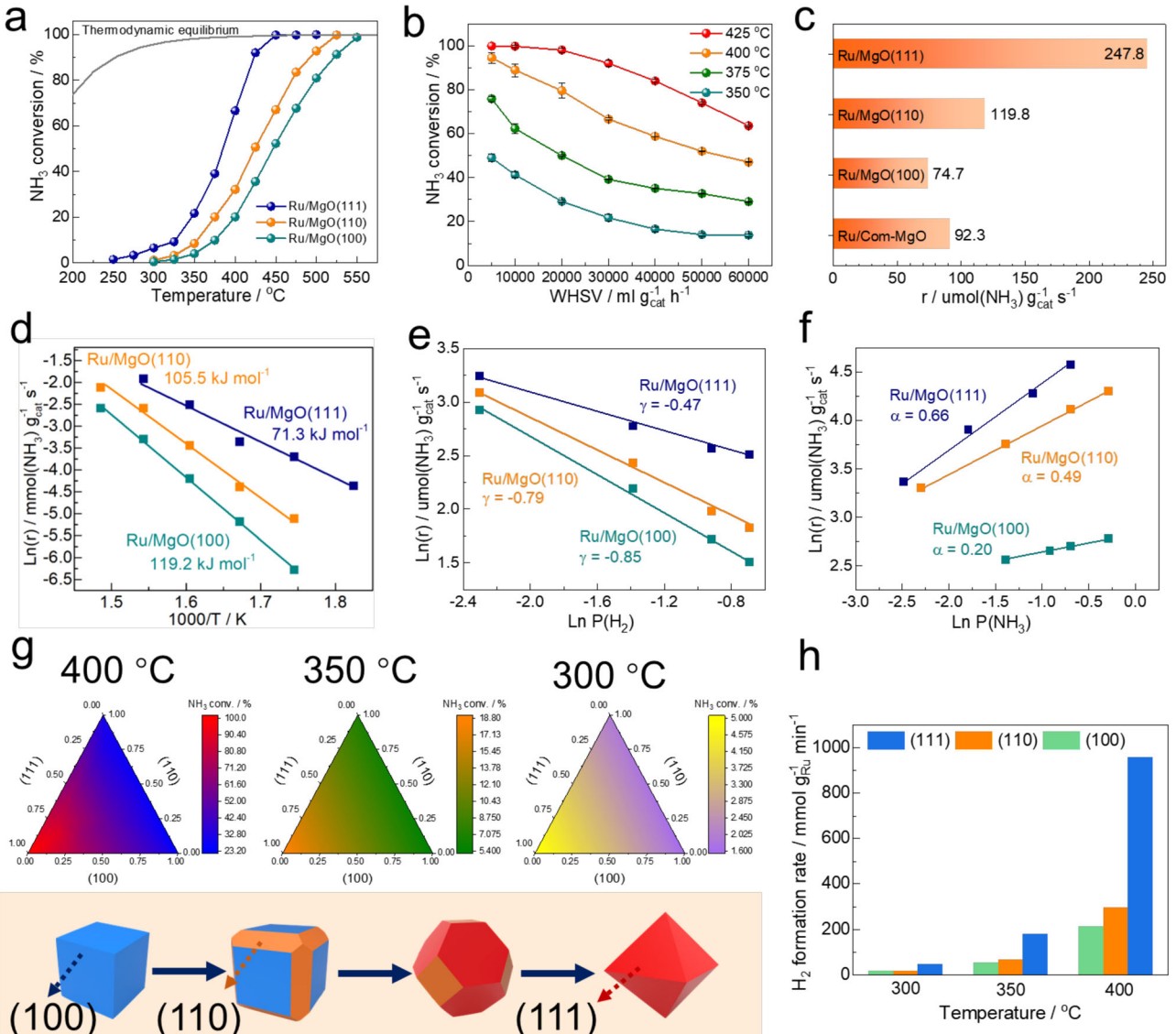

**Fig. 2 | Catalytic performance tests for ammonia decomposition. a** $NH_3$ conversion as a function of reaction temperature over different supported Ru samples, reaction conditions: weight hour space velocity (WHSV) = 30,000 mL $g_{cat}^{-1}$ $h^{-1}$, 1 bar; **b** effect of WHSV on the $NH_3$ conversion of Ru/MgO(111) at different temperatures; the error bars in the figure represents standard deviation through three repeated measurements; **c** reaction rate for ammonia decomposition over Ru-based catalysts with different MgO supports, reaction conditions: T = 400 °C, WHSV = 30,000 mL $g_{cat}^{-1}$ $h^{-1}$, 1 bar; **d** Arrhenius plots of supported Ru samples;

dependence of ammonia decomposition rate of supported Ru samples on the partial pressures of (**e**) $H_2$ and (**f**) $NH_3$ at 350 °C. α and γ represent the reaction order of $NH_3$ and $H_2$, respectively. **g** 2D contour maps of activity over the supported Ru samples with different (111), (110) and (100) formulations at 400, 350 and 300 °C, reaction conditions: WHSV = 30,000 mL $g_{cat}^{-1}$ $h^{-1}$, 1 bar. The bottom scheme shows the illustration of reconstruction of MgO-cube to MgO-octahedron; **h** comparison of $H_2$ formation rate over different exposed MgO facets based on the analysis of 2D contour maps; the metal loading was based on the ICP-OES results.

heavier Ru atoms. Importantly, the line mapping images in Fig. S5d–f acquired along the lines from Fig. S5a–c confirm that Ru atoms in the atomic column are bonding with O atoms, sitting on the hollow sites of 3 O atoms, referred to Ru-OOO site, in agreement with XPS analysis. Further X-ray absorption fine structure (EXAFS) was performed to investigate the average coordination environment of Ru species. Significant contribution from Ru–O bond was observed in Fig. 1g and subsequent least square fitting reveals a 3-oxygen coordination to the Ru at 2.03(2)Å with a distance of 3.17(3)Å from Mg (Table S2), concomitantly matching with the STEM analysis and atomic model (Fig. S3 and Fig. 1g-insert) but there is no Ru–Ru bond at all. In contrast, the EXAFS spectrum for Ru/MgO(110) only show a 6 coordination Ru–Ru bond without any contribution from Ru–O (Fig. S6 and Table S2).

These results reveal that the atomically-dispersed Ru/MgO(111) with precise Ru-OOO hollow sites can be successfully synthesized.

**Catalytic performance and analysis for ammonia decomposition**
The catalytic performance of as-built hierological Ru/MgO catalysts was directly evaluated for the decomposition of ammonia under low temperatures. The ammonia decomposition is a mildly endothermic process that involves successive cleavage of N–H bonds and recombination of N and H atoms to form $N_2$ and $H_2$. Although it begins to decompose at low temperatures as shown in Fig. S7, the equilibrium conversion of ammonia is 98–99% at 425 °C and thus the practical conversion is highly dependent on both catalysts and temperatures used. Figure 2a shows the temperature dependence for ammonia conversion over the Ru/MgO catalysts. It can be seen that the catalytic

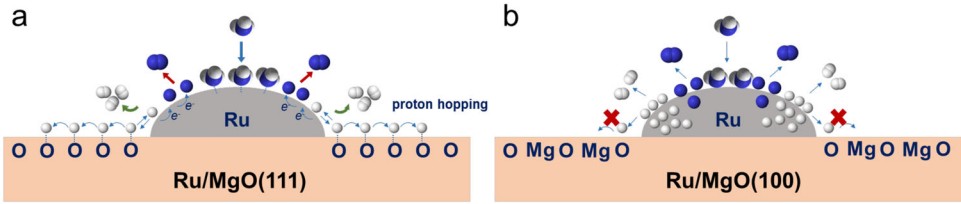

**Fig. 3 | Mechanism illustration of ammonia decomposition on Ru/MgO(111) and Ru/MgO(100).** The different behaviors of hydrogen hopping in (**a**) Ru/MgO(111) and (**b**) Ru/MgO(100).

performance shows a S-type curve and the conversion increases with the increase of temperature over these catalysts. Notably, the Ru/MgO(111) exhibits higher activity than those of Ru/MgO(110) and (100) and achieves close to the thermodynamic equilibrium at 425 °C with a high WHSV of 30,000 mL $g_{cat}^{-1}$ $h^{-1}$, manifesting the importance of surface property of MgO(111) and Ru chemical states. The effect of WHSV was further conducted. Figure 2b depicts that the $NH_3$ conversion decreases with the increase of WHSV, which is due to the shorter residence time for $NH_3$ absorbed on active sites. Figure 2c carefully compares the reaction rates at 400 °C over these Ru/MgO catalysts. The Ru/MgO(111) exhibits a superiority in reaction rate with 247.8 μmol($NH_3$) $g_{cat}^{-1}$ $s^{-1}$ at 400 °C, which is about 2.1 times and 3.3 times higher than those of Ru/MgO(110) and (100), respectively, which do not follow their surface area value (see Table S3). It is worth noting that the TOF value of Ru/MgO(111) is 2.33 $s^{-1}$ at 400 °C, higher than that of Ru/MgO(110) (0.90 $s^{-1}$) and Ru/MgO(100) (0.59 $s^{-1}$), manifesting the importance of exposed MgO facets on catalytic performance. The Ru supported on commercially available MgO sample (Ru/Com-MgO) displays a comparable reaction rate between Ru/MgO(100) and Ru/MgO(110). This finding reveals that the Com-MgO is predominantly composed of more stable (100) and (110) facets, which has lower surface energy.

Kinetic studies were performed to depict the peculiar effect of using polar MgO(111) as support. Arrhenius plots were firstly obtained by linear fitting of Ln (reaction rate) versus 1/T (Fig. 2d). The apparent activation energy (Ea) estimated from this plot corresponds to the Ea in the rate-determining step (RDS) in the overall chemical process, which is always considered to be N recombination[26,49]. Ru/MgO(110) and Ru/MgO(100) without significant exposure of polar surfaces exhibit Ea values of 105.5 and 119.2 kJ $mol^{-1}$, which are in agreement with normal 110–130 kJ $mol^{-1}$ over traditional Ru/MgO catalysts[50–52]. The Ea was significantly reduced to 71.3 kJ $mol^{-1}$ over Ru/MgO(111). This finding posts the important nature of polar surface on promoting the chemical process of ammonia decomposition. For further understanding, the reaction orders were analyzed to obtain information on the rate-determining step and the poisoning species in ammonia decomposition (Table S4). The partial pressure of $N_2$ was found to have negligible effect on the reaction rate over all Ru-based catalysts (Fig. S8). These results suggest that the N recombination is the RDS, in agreement with previous conclusions[26,53,54]. As shown in Fig. 2e, the hydrogen orders (γ) for both Ru/MgO(110) and Ru/MgO(100) were large and negative, reaching −0.79 and −0.85, respectively. This indicates that hydrogen adatoms are strongly adsorbed onto the Ru surface, well-known as hydrogen poisoning, preventing further activation and chemisorption of ammonia and nitrogen species on these competitive Ru sites. Surprisingly, the hydrogen order of Ru/MgO(111) is only −0.47, smaller than that of Ru/MgO(110) and Ru/MgO(100). This finding indicates an effective hydrogen removal rate on the Ru surface in Ru/MgO(111). The ammonia order (α) for both Ru/MgO(110) and Ru/MgO(100) was small and positive, which reveals an obvious retardation of nitrogen species, such as N and NH species on the Ru surface (Fig. 2f). The small α value indicates that the nitrogen desorption step was kinetically slow. It is worth noting that the α exhibits larger value of 0.66

over the Ru/MgO(111). It is believed that the strong affinity from terminal oxygen anions on the polar surface of non-reducible MgO(111) allows the proton spillover from Ru surface to $O^{2-}$ covered surface in the cracking N–H bond, and retained electrons at Ru species are beneficial for back donation to Ru–N antibonding orbital, reducing the N retardation and promoting the rate of N–N recombination. Compared with non-polar Ru/MgO(110) and Ru/MgO(100), the small−γ/α value over Ru/MgO(111) indicates that hydrogen adatoms do not block the active sites and the nitrogen species desorb more easily from the catalyst surface (Table S4). Figure 3 is depicted to further illustrate the varied reaction order between different MgO supported Ru catalysts. Compared with MgO(100), the exposed MgO(111) surface exhibits a strong affinity for hydrogen activation (via Frustrated Lewis pair with Ru species) and removal (proton hopping), corresponding to its smaller γ value, which helps to reduce the hydrogen poisoning. In addition, the reduction of Ru during hydrogen activation can also promote the N–N recombination by donating the electrons to the Ru–N antibonding orbitals, corresponding to the large α value.

It is worthwhile to note that despite Ru/MgO(111) with predominant exposure of the (111) facet, there are still a considerable amount of (110) and (100) facet in the same sample. For approaching quantitative analysis of individual contribution of the (111), (110) and (100) facet towards the ammonia decomposition rates, surface reconstruction strategy was employed to cleave MgO from cube to octahedron shape, see experimental for more details[55–57]. The MgO cube exposes all (100) facets and the cleavage allows the formation of (110) and then (111) facets. Thus, a series of Ru/MgO catalysts with different ratios of (100), (110) and (111) facets were successfully synthesized (Fig. S9). Figure 2g demonstrates the catalytic performance for ammonia decomposition over these catalysts at 400, 350 and 300 °C. The contour maps show how the $NH_3$ conversion (intensity) varies as a function of chemical formulations, based on the analysis of exposed facets. It can be seen that (111) facet displays the highest performance in all cases, followed by that of (110) and (100) facets, based on the analysis between the activity and facet percentages. The promoting effect of polar (111) facet is superior at higher reaction temperatures. Figure 2h summarizes contributions of different facets on $H_2$ formation rates. Significantly, a 4.3-fold and 3.4-fold higher $H_2$ formation rates were observed over the Ru supported on (111) facets than that of (100) and (110) facets even under low reaction temperature of 400 °C. These results post the importance of polar surface for efficient ammonia decomposition.

To verify the N–H and Ru–N activation as well as the hydrogen removal ability of the polar MgO(111) demonstrated in the above kinetic studies, a range of dynamic characterization has been performed. Temperature-programmed surface reaction (TPSR) coupled with mass spectrometry (MS) was first employed to investigate the ammonia activation process, as shown in Fig. 4a. After pre-activation of the catalysts, $NH_3$ gas was introduced into the reactor at 50 °C for 30 min for $NH_3$ chemisorption on the catalyst surface. Argon was used as the purge gas and the temperature was increased to 800 °C at 5 °C/min. The production of $H_2$ and $N_2$ was immediately observed with the decreasing signal of $NH_3$ for both Ru/MgO(111) and Ru/MgO(100)

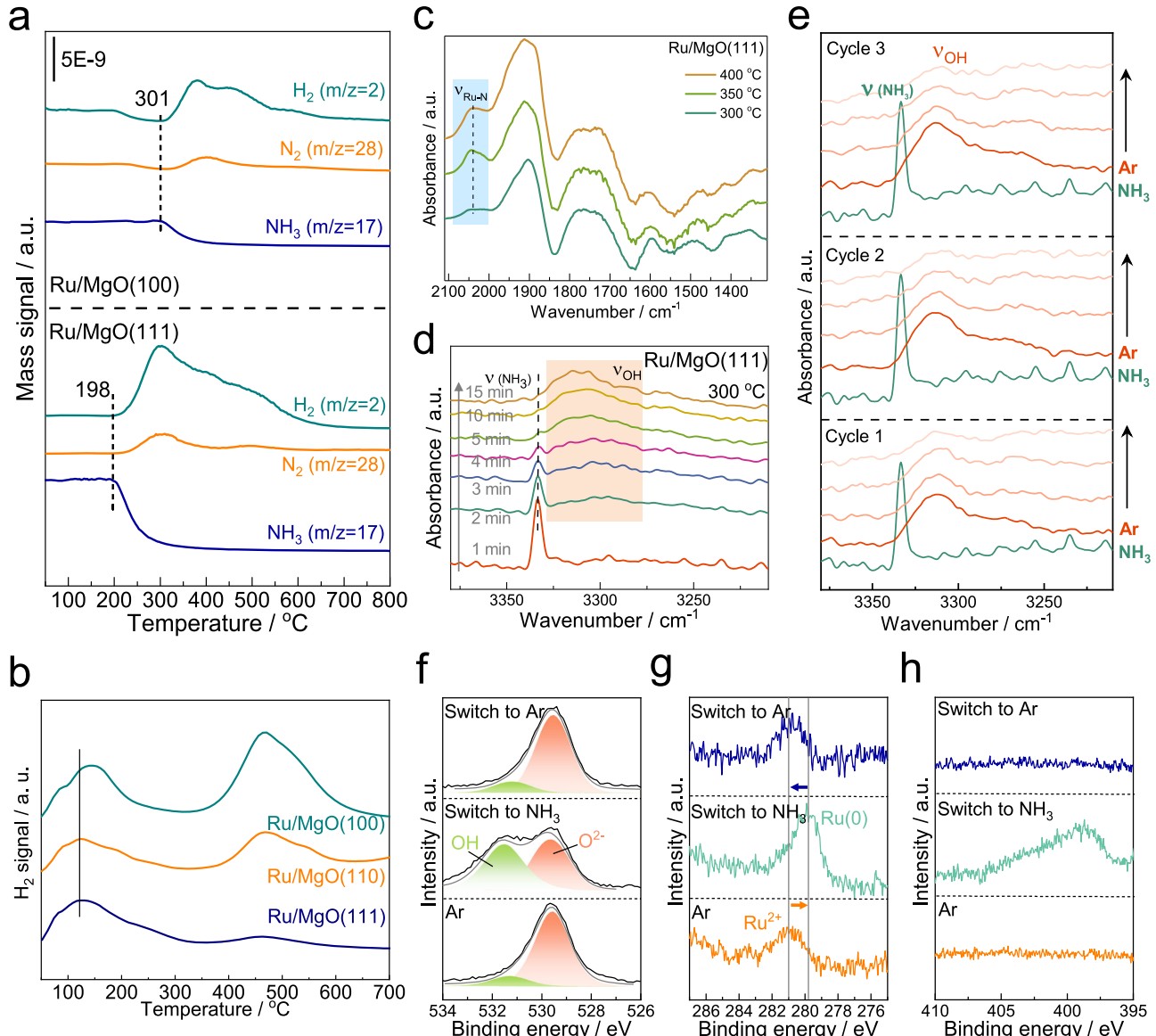

**Fig. 4 | Characterizations of verifying the effect of hydrogen hopping in ammonia decomposition. a** TPSR over the Ru/MgO(111) and Ru/MgO(100) based on the NH$_3$ decomposition reaction; **b** H$_2$-TPD profiles of supported Ru samples; **c** in situ DRIFTS study of NH$_3$ adsorption over Ru/MgO(111) under different temperatures; **d** time-resolved DRIFTS of NH$_3$ transformation as a function of time over Ru/MgO(111) at 300 °C; **e** reversible cycles of DRIFTS for NH$_3$ transformation under alternative sweeping between NH$_3$ and Ar at at 300 °C; **f** O 1$s$, **g** Ru 3$d$ and **h** N 1$s$ of AP-XPS spectra at 300 °C over Ru/MgO(111) under alternative sweeping between 0.9 mbar Ar and 0.9 mbar NH$_3$.

samples. Importantly, the decrease of NH$_3$ signal over Ru/MgO(111) occurs at 198 °C while the NH$_3$ starts to decompose at about 300 °C over Ru/MgO(100), indicating the superior activation of NH$_3$ on the interface of Ru and MgO(111) surface. These results correspond to the lower Ea over Ru/MgO(111). Besides, the larger signals of detected H$_2$ and N$_2$ reveal higher capability for ammonia adsorption which is ascribed to the highly dispersion of active sites. Further H$_2$-TPD with MS detector was investigated for understanding of the chemisorption behavior of the hydrogen species, as shown in Fig. 4b. It can be seen that a broad desorption peak can be recognized in all samples at about 125 °C, which is ascribed to the weak adsorption of hydrogen species. Another H$_2$ desorption peaks located at about 475 °C was observed over the Ru/MgO(110) and Ru/MgO(100) samples, corresponding to the strong adsorption of hydrogen. That is to say, hydrogen species are strongly bonding on the surface of Ru and results in severe hydrogen poison. On the contrary, the higher temperature desorption peak for Ru/MgO(111) has a significantly lower intensity relative to the lower-

temperature peak at 125 °C and there is an additional shoulder peak at around 200 °C. This might again be indicative of the reversible transfer of surface H-species from Ru onto MgO(111) as demonstrated by aforementioned TPSR measurement. As a result, the adsorbed hydrogen atoms could be quickly transferred away from the Ru active sites once they are dissociated, which agrees coherently with the H reaction order kinetic study. This result reveals that the hydrogen species are more feasible to be removed from the Ru surface.

The direct observation of the intrinsic species during NH$_3$ decomposition is of great importance to understand the nature of reaction mechanism. Thus, the in-situ DRIFTS analysis for ammonia transformation was employed. Figure 4c shows the evolution of principal surface species for ammonia decomposition under different reaction temperatures. The Ru−N species (2047 cm$^{-1}$) was immediately observed on Ru/MgO(111) and the signal intensity increase as the temperature increase from 300 to 350 °C and then levels off at 400 °C[58,59]. The lower Ru−N band intensity at 300 °C might not

necessarily mean a drop in catalytic performance, instead, the adsorbed ammonia is swiftly activated and recombined with another N atom to form dinitrogen. Additionally, this can be attributed to the circumvention of kinetic barrier and hence enhancement of the ammonia adsorption, similar to the phenomenon observed in the temperature and WHSV dependence experiments. Instead of the observation of on-top Ru–H at 1802 cm$^{-1}$, the peak at 1900 cm$^{-1}$ assigned to the bending mode of [OH] was clearly seen; simultaneously, the intensity of the [OH] stretching band at 3300 cm$^{-1}$ is also found to rise as the temperature increases (Fig. S10)[18,60]. This signifies the incessant transfer of hydrogen atoms from Ru to the MgO(111) support as protons during the activation of the adsorbed ammonia, in consistent with the above kinetic analysis with a small hydrogen reaction order. In other words, it is the facile removal of hydrogen from Ru active sites to support from this strong metal support interaction that maintains the robust ammonia activation.

In contrast, the observation of sharp intermediate NH$_2$ species at 1573 cm$^{-1}$ together with relatively smaller intensity of Ru–N peaks for Ru/MgO(100) implies the sluggish dehydrogenation/recombination of ammonia to form the end products of atomic nitrogen and hydrogen (Fig. S11)[18]. The absence of [OH] stretching region implies that H spillover cannot take place on Ru/MgO(100) from the Fig. S12.

Time-resolved DRIFTS was then performed to disseminate the relationship between NH$_3$ uptake and the corresponding surface change of the MgO support, as shown in Fig. 4d. Initially, a strong signal of NH$_3$ appears at 3333 cm$^{-1}$ assigned to the N–H stretching with the introduction of NH$_3$ gas. Notably, an increasing absorbance of a broad peak at about 3300 cm$^{-1}$ was found, which is attributed to the [OH] stretching region for Ru/MgO(111). From the kinetic point of view, it is impossible for MgO(111) itself to dissociate hydrogen, which implies the growth of surface [OH] is presumably be originated from the transfer of hydrogenic species from Ru during the ammonia decomposition. This finding implies that atomic H on Ru could be spilled favorably to the oxygen-terminated surface of MgO(111) to form protons (as hydroxyl species), correlated with the strong H removal ability of the support.

To further confirm the reversible change of the [OH] species on the Ru/MgO(111) surface, the DRIFTS was further performed where the sample feed was dynamically switched between ammonia and argon in operando, as evident in Fig. 4e. The ammonia was first introduced into the chamber and heated to 300 °C. The rapid formation of [OH] species with the disappearance of NH$_3$ signal was found (the red line in cycle 1). Upon passing argon, the [OH] peak diminishes, indicating that the migration of the proton on the oxygen-terminated surface is reversible, leading to spontaneous return onto vacant Ru to form H adatom which eventually recombines with another H adatom and desorbs as dihydrogen molecule. These results reveal that polar MgO(111) surface allows the protons transfer from Ru surface to bond with O$^{2-}$ during the decomposition of ammonia and these protons can also be reversibly removed from the MgO(111) surface for dihydrogen production, which prevents the hydrogen poisoning and facilitates the efficient decomposition of ammonia. When no further change is observed for the [OH] region, the gas feed was switched back to ammonia for second cycle measurement. The shape, peak position, and intensity of the peaks are in general similar to the first cycle, implying the reversibility of such hydrogen spillover process to maintain effective H removal to alleviate H poisoning and subsequent migration to form end-product dihydrogen. The third-cycle confirms the stability of reversible transfer of protons. All-in-all, this cycle-test has proved the sustainability of the spillover phenomenon to facilitate ammonia decomposition.

It is also essential to investigate the dynamic change of the oxidation state of Ru and surface oxygen species during the ammonia decomposition. AP-XPS was performed at an elevated temperature of 300 °C under dynamic switching between Ar (0.9 mbar) and

NH$_3$ (0.9 mbar), as shown in Fig. 4f–h. Initially, the Ru on MgO(111) demonstrates an oxidized state of +2 under a non-reducing atmosphere. The O 1$s$ spectrum shows a peak ascribed to the typical oxide with trace [OH] species. After switching the sample gaseous feed to ammonia, the binding energy of Ru shifts to lower energy from 281.1 to 279.9 eV, indicating the reduction of Ru$^{2+}$ to Ru$^0$ (Fig. 4g). Notably, a pronounced but simultaneously increase of OH$^-$ species formed as shown in Fig. 4f. This is due to the cracking of NH$_3$ on the Ru atoms to give protons coverage on surface O$^{2-}$ of MgO(111) during ammonia decomposition. The retained electrons enrich the electron density of Ru and then render the activation of Ru–N bond by donating electrons to the antibonding orbital, which allows the atomically-dispersed Ru$^{2+}$ to exhibit high efficiency on ammonia decomposition. The surface N species is also visible from the N 1$s$ profile (Fig. 4h). Interestingly, the reduced Ru$^0$ state swiftly swings to Ru$^{2+}$ state and the [OH] species decrease again when the gas flow is switched to Ar. It is envisaged that the hydrogenic species capture electrons from Ru inversely to produce dihydrogen. As expected, the N species disappear after Ar purging, indicating a totally removal of N atoms from Ru surface to produce dinitrogen. It is believed that the exceptional activity for NH$_3$ dehydrogenation over Ru/MgO(111) with surface Ru$^{2+}$ with terminal O$^{2-}$ species of MgO(111) is also consistent with the heterolytic activation of ammonia over the mechanism of Frustrated Lewis Pair, FLP (as Ru$^{2+}$–O$^{2-}$)[61], which explains the fact that such simple 'NaCl' structure of MgO can exert such significant support effect for Ru to this reaction.

On the contrary, there is neither a shift in binding energy for Ru 3$d$ nor shape changes in the O 1$s$ profiles of Ru/MgO(100) (Figs. S13 and S14). Therefore, we found an obvious hydrogen spillover from Ru to O-terminated surface of polar MgO(111) initiated by FLP and hence proton hopping across the surface O sites. This phenomenon ensures the heterolytic cleavage of N–H bonds to prevent hydrogen poisoning on the Ru surface.

Given that the polar MgO(111) exhibits efficient reversible hydrogen migration (hopping) with atomic Ru incorporation, it is significantly important to study the synergy of Ru and the support at progressive increase in surface Ru concentration[20,61]. It has been well established that ammonia decomposition is a structure sensitivity reaction and simple B$_5$ step sites on Ru NPs surface are responsible for the activity of this reaction. These sites are composed of three atoms in one layer and two additional atoms above this at a monoatomic step on an Ru(0001) terrace, which could provide appropriate geometric orbitals for reversibly cleavage/construction of N$_2$. Previous reports have shown that the activity of ammonia decomposition depends on the concentration of B$_5$ sites and isolated single-atom sites perhaps without strong metal support interaction are not efficient for this reaction[8–12]. Interestingly, our Ru/MgO(111) (at below ca. 3.1 wt.% Ru loading) displays an atomic dispersion of adsorbed Ru without any sign for the formation of B$_5$ sites or other Ru clusters as shown by STEM (Figs. 1d, e and S3) and EXAFS (no Ru–Ru bond as shown in Figs. 1g and S15), which could not form this type of coordinated stepped sites. Nevertheless, it clearly exhibits superior catalytic performance for ammonia decomposition in our case. To gain further insight, Ru/MgO(111) with different Ru loadings were carefully prepared and evaluated for ammonia decomposition. Figure 5a shows the NH$_3$ conversion rate as a function of Ru loading from 0.01 wt.% to 7.0 wt.%. As expected, the Ru species display an atomic dispersion when the Ru loading is below 3 wt.% (Fig. 5b). It is interesting to note that the reaction rate demonstrates a double volcano-type trend. At low Ru loading, the TOF value rapidly increases with increase of the surface Ru loading, achieving the maximum TOF value of 3.43 s$^{-1}$ and 4.91 s$^{-1}$ at 400 and 450 °C, respectively. In this case, the distance between Ru atoms is getting shorter and the Ru concentration on

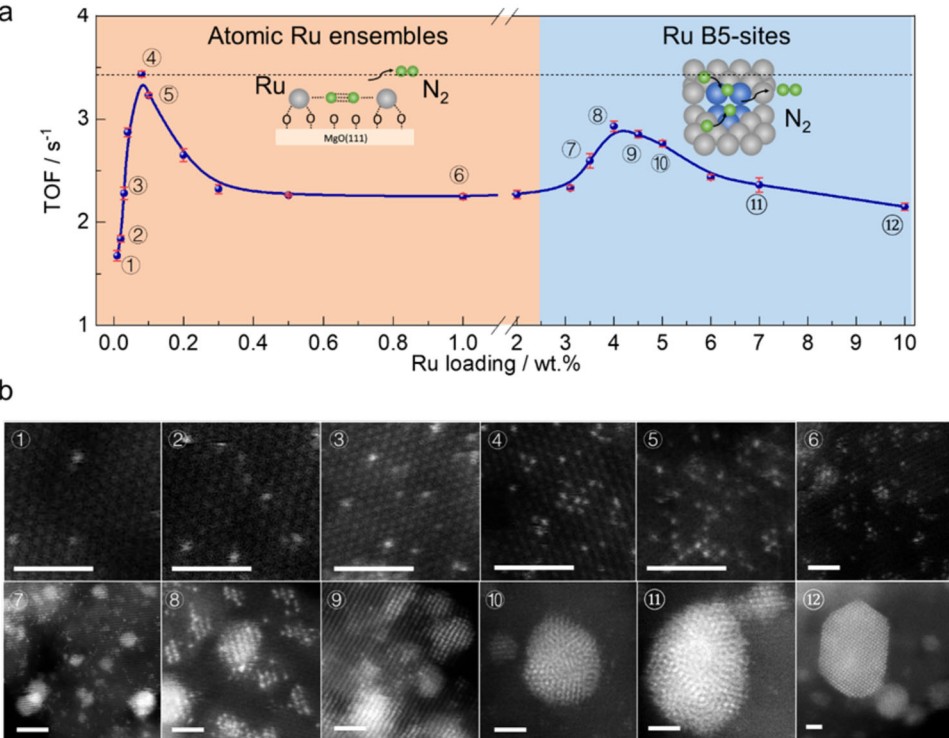

**Fig. 5 | Loading effect of Ru from atomic form to cluster/nanoparticle on MgO(111). a** The TOF values with error bars for ammonia decomposition as a function of Ru loading at 400 °C; the error bars in the figure represents standard deviation through three repeated measurements; **b** typical STEM images of Ru/MgO(111) with different loadings, the labeled numbers are corresponding to the samples in **a**; the scale bar in white is 1 nm.

MgO(111) surface becomes higher to form surface sites in vicinity such as 2×Ru atoms and 3×Ru atoms as the Ru loading increases (Fig. 5b ①–⑥). That is to say, the neighbor surface Ru atoms exhibit a synergistic effect on the $NH_3$ decomposition to enhance the efficiency of single-atom Ru species, as similar to forward $N_2$ reduction reaction[62,63]. It is worth noting that the atomically dispersed Ru/MgO(110) and Ru/MgO(100) with an ultra-low loading of 0.01 wt.% exhibit negligible activities compared with Ru/MgO(111) with the same loading (Table S5), which reveals that the superior performance for ammonia decomposition over the atomically dispersed Ru/MgO(111) is due to the synergistic effect between the Ru and surface O atoms. As we discussed above, the MgO(111) surface displays a superior ability for hydrogen hopping during the N–H bond cleavage to prevent hydrogen poison and the retained N species on Ru surface can be removed via formation of $N_2$. It can be suggested that the close-by neighbor Ru atoms allow the feasible N–N bond recombination and thus improve the efficiency of $NH_3$ decomposition. However, further increasing the Ru loading beyond the surface dispersion leads the formation of clusters and nanoparticles. Another volcano-type curve can also be seen clearly in Fig. 5a on the right side. This demonstrates a strong particle size effect on the forward reaction rate (Fig. 5b ⑦–⑫). The TOF value increase to $2.90\,s^{-1}$ when the particle size is about 2 nm, which is in agreement with the presence of highest $B_5$ sites on these Ru NPs, corresponding to previous studies[14,16]. Larger metal particles with larger surface coverage to the support exhibit lower intrinsic activity on which would cause a sharp lost in effective utilization of noble metal. It is worth noting that the efficiency over these samples with $B_5$ sites is still lower than most of atomically-dispersed samples, presumably ammonia activation is by the typical homolytic dehydrogenation rather than 'heterolytic' means in the case of $Ru^{2+}$–$O^{2-}$. Therefore, the atomically-dispersed Ru with close interconnections on the polar MgO(111) appears to be a superior

configuration, which shows a more efficient catalytic performance than the classical stepped sites for ammonia decomposition to achieve the maximum utilization of noble metal.

## DFT calculations and mechanism

The above activity-loading relationship at the atomic dispersion (Fig. 6a) suggests that the reaction is likely to proceed via the formation of surface Ru–'N' pairs or even higher Ru surface 'oligomers' in a close proximity. As a result, a computational model of simple 2× Ru atoms on a 5×5 supercell MgO(111) was constructed. The initial position of the adsorbed Ru atom was based on the 3-oxygen coordination model described above. This structure was theoretically confirmed to be the most stable model for Ru adsorption and is more stable than embedding Ru in place of a subsurface Mg atom (Fig. S16a). The Bader charge of Ru (1.5|e|) also agrees well with experimental observation of the oxidation state of +2. A stoichiometric MgO(111) model was presently considered but the calculated adsorption energy of Ru on stoichiometric and non-stoichiometric MgO(111) was found similar (–13.35 eV vs –12.95 eV) (Fig. S16b), in agreement with our previous results[64]. It is clear from the modelling that the Ru does not transfer charge to the opposite side of the slab. Instead, the electron density from Ru is donated to and shared by the surface O atoms in the periphery of Ru (Fig. S16c). This result agrees well with our previous results[20,61], and our current experimental XPS results.

Another Ru atom was then placed on the MgO(111) with the optimized Ru–Ru distance determined to be 5.04 Å as the close by neighbor pair in this geometric model (Fig. S17). This atomic distance is in line with no detection of any short-range Ru–Ru bond in metallic aggregate from EXAFS and with the distance observed between two Ru atoms in STEM image (Fig. S3). Two ammonia molecules were then introduced to the system where they were allowed to adsorb on the Ru until the adsorption geometries were optimized. Similar atomic Ru pairs supported on MgO(110) and MgO(100) models were also constructed for comparison (Fig S17). The ammonia molecules were found

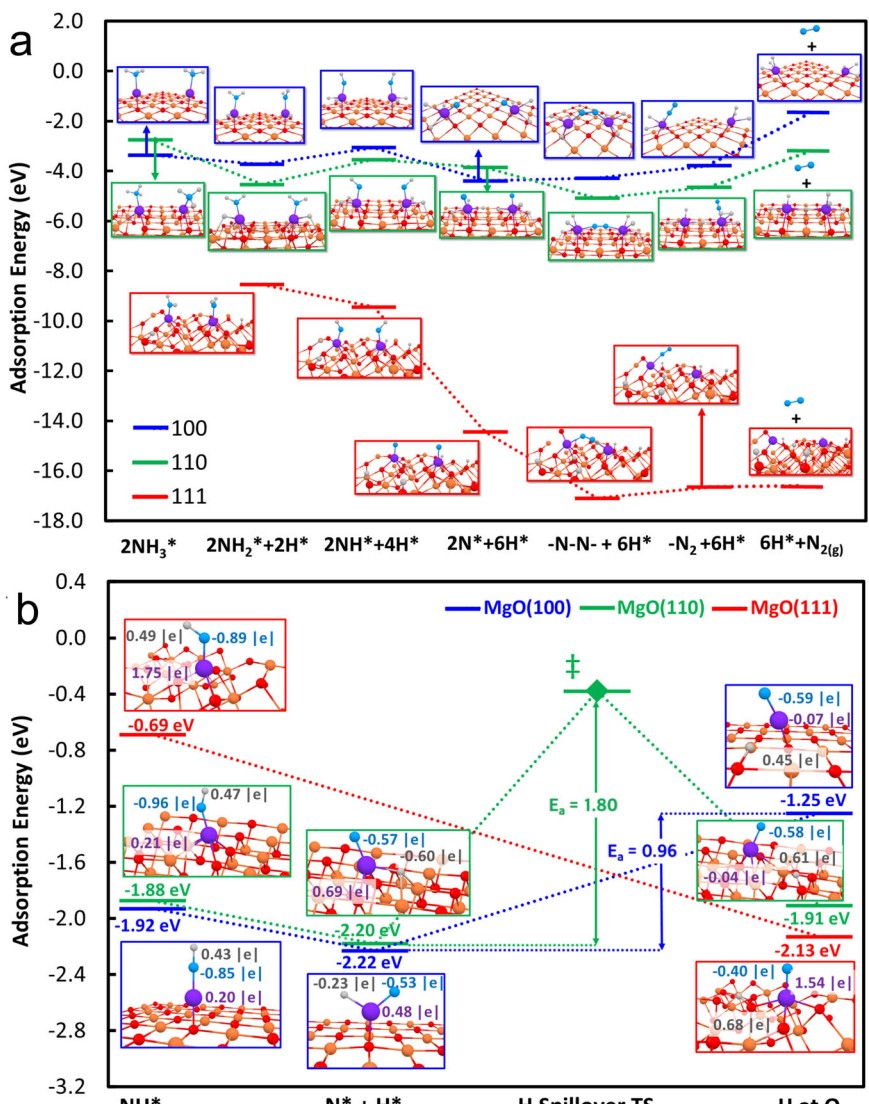

**Fig. 6 | DFT calculations of Ru/MgO(100), Ru/MgO(110) and Ru/MgO(111).** Energy profiles and their corresponding optimized intermediate structures for (**a**) ammonia decomposition and N–N recombination pathways to form $N_2$ at MgO-supported two Ru single atoms and (**b**) hydrogen removal from NH* species and hydrogen spillover at MgO-supported Ru single atom. The reactions were calculated over (100), (110) and (111) facet of MgO shown in blue, green and red lines, respectively. $E_a$ (in eV) refers to the activation energies of hydrogen spillover from the NH* species to the three MgO supports. The optimized structure of each intermediate is depicted wherein Mg, O, Ru, N and H atoms are depicted as orange, red, purple, blue and gray spheres, respectively; the values near the atoms of H, N and Ru in the inserted structure are their respective Bader charges (in |e|). The value near each intermediate refers to their corresponding adsorption energy (in eV), with respect to a single $H_2$ and $N_2$ gas.

to undergo stepwise dehydrogenation where the nitride species interact for recombination to form dinitrogen.

Figure 6a shows the calculated energy profiles for the stepwise dehydrogenation of two $NH_3$ molecules to $N_2/H_2$ over this simple binuclear model where two surface Ru single atoms (2× Ru) in close proximity over MgO(111). Experiments suggest that the two $NH_3$ molecules do not seem to adsorb only on a single Ru at the same time but on two separate atomic Ru sites. This suggests recombination of $N_2$ is unlikely catalyzed by a single metal site, as similar to the findings in homogeneous counterparts[28]. Thus, a similar mechanism was assumed in the DFT elucidation of the $NH_3$ decomposition pathway. Interestingly, stepwise dehydrogenation of ammonia is energetically favorable to take place as experimental data indicated, due to the strong affinity of H on polar O terminal MgO(111) in vicinity by electrostatic means, as previously discussed. Apparently, the final N–N recombination to $N_2$ involves the synergetic coupling of two Ru≡N species on both neighbored surface metal sites in the close proximity without going to the

pathways using single metal site or complex $B_5$ in metal nanoparticle. Overall, the calculated activation energies of 0.19 to 1.21 eV are required (Fig. S18). In $N_2$ recombination process, the barrier is the highest (1.21 eV) among all steps, which is consistent to the observed activation process (Fig. 2d) in rate determining step of the reaction. Given the simultaneous production of $H_2$ and $N_2$ without time-lag as obtained in TPSR, it is anticipated that dynamic transfer of H onto the MgO(111) also assists the kinetics of dehydrogenation of ammonia by fast H migration from Ru metal to this particular polar support surface and to other metal sites for $H_2$ recombination.

We further compared the complete $NH_3$ decomposition reaction coordinate using models of $(NH_3-Ru)_2$ on MgO (100), (110) and (111) in Fig. 6a. On 2Ru/MgO(100) and 2Ru/MgO(110), initial H removal from adsorbed $NH_3$ species are unfavorable. Furthermore, the H atoms, unable to spillover to MgO, crowd the atomically dispersed Ru catalyst. In contrast, the adsorption of $NH_3$ on 2Ru/MgO(111) immediately leads to removal of H atoms and their

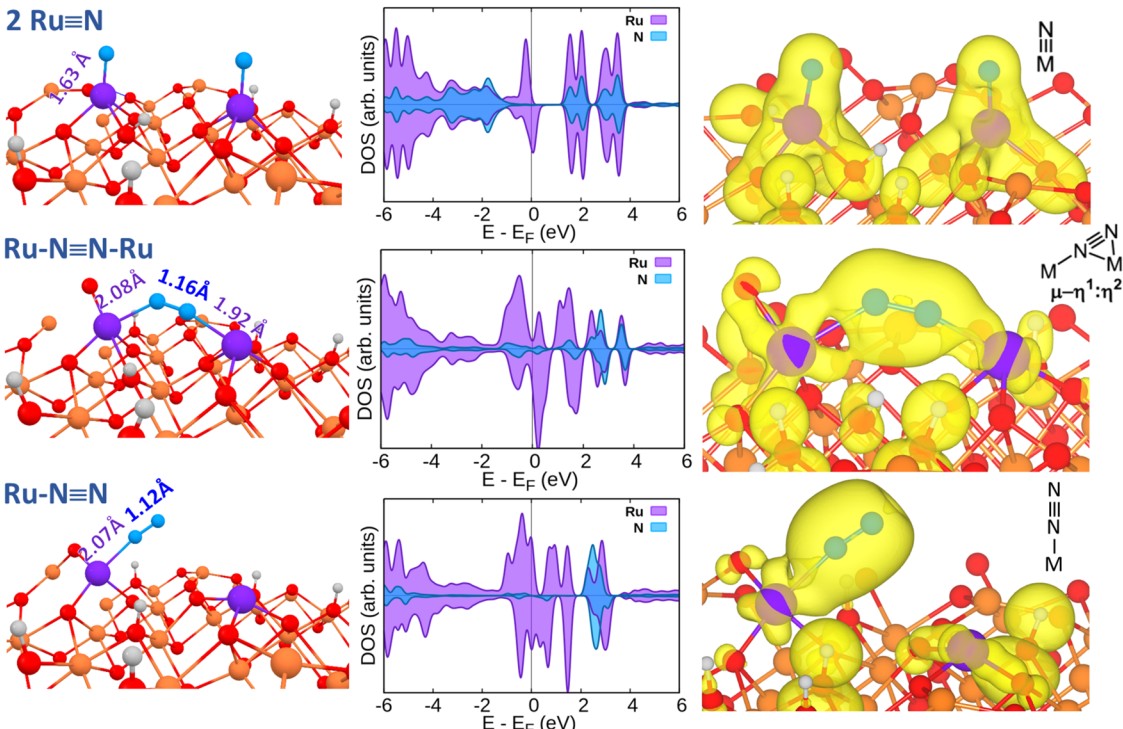

**Fig. 7 | Optimized structures, partial density of state and partial charge density of intermediate on Ru/MgO(111).** Ammonia decomposition starting from 2 surface Ru≡N to N₂ *via* μ-η¹:η² bimetallic nitrido-species complex like structure in homogeneous systems[28,62,64] as intermediate.

migration to the MgO surface with a barrierless process. Subsequent removal of H atoms from $NH_x$ species are also thermodynamically favorable on $2Ru_2/MgO(111)$ due to absence of crowding of the Ru catalyst and abundance of surface O atoms where H can migrate. We note that the removal of H from catalyst surface may not affect the rate-determining step of the reaction[64]. However, our experimental results show that the hydrogen migration step (avoid poisoning) is crucial in the overall kinetics when much milder conditions are used. As we have showed that at low pressures, which is more relevant for renewable energy transport, the support interaction becomes important. The use of polar MgO(111) surface could render the kinetics much less dependent on hydrogen (spillover readily achieved) under the milder conditions; on the other hand, hydrogen adatoms are strongly adsorbed onto the Ru surface, well-known as hydrogen poisoning, was further observed on both Ru/MgO(110) and Ru/MgO(100) (Fig. 2e), preventing further activation and chemisorption of ammonia and nitrogen species on these competitive Ru sites. To confirm the unique hydrogen spillover phenomenon on Ru/MgO(111), we analyzed the case of the removal of H from *NH species adsorbed on a single Ru atom supported on MgO(100), MgO(110) and MgO(111), see Fig. 6b. On Ru/MgO(100) and Ru/MgO(110), H atoms removed from the *NH species, remain on the Ru surface as their migration to the support surface is not energetically favorable and with barriers of around 1-2 eV. In contrast, the *NH on Ru/MgO(111) interacts with the surface O atoms of MgO in the periphery of Ru. This leads to the cleavage of the N−H bond wherein the H atoms move spontaneously to the surface O atoms of the support with a barrierless process. Thus, on Ru/MgO(111), the reduced H poisoning observed in the kinetic experiments is explained by the favorable spillover of H to the MgO surface. Furthermore, analysis of the Bader charges shows that the spillover H donates electron density to Ru (Fig. S16c), consistent with the simultaneous reduction of Ru and increase of surface OH⁻ species observed in the AP-XPS analysis. It is believed that spillover H will likely go back to Ru sites to acquire electrons and produce H₂ but this process is fast and does not affect the

experimentally observed ammonia decomposition activity. Thus, to avoid distraction, we have excluded the H₂ desorption step in the DFT calculations. Instead, we have chosen to account for the effect of the unremoved of H* atoms on Ru for 2Ru/MgO(100) and 2Ru/MgO(110) and spillover H−O on Ru₂/MgO(111) on the thermodynamics of the steps toward N−N recombination. We also examine the desorption of N₂ from the surfaces in Fig. 5a, it is obvious to see that the breakage of the last Ru·N bond to form N₂ is much more thermodynamically favorable in MgO(111), compared to the other non-polar (100) and (110) facets. In short summary, the current DFT work provides consistent structural, energetic and electronic structure properties that agree with experimental data.

To gain more insight into the detail in the N₂ combination steps, we also analyze their partial density of state (PDOS) of these intermediates (Fig. 7). As we can see that a complete overlap between 3d orbital electrons of Ru with 2p orbital electrons of N for two single N atoms bound to two single Ru atoms to form Ru≡N of typical triple bond distance of 1.63 Å (cf. 1.64Å[62]) in range of 0 eV to −6 eV with respect to the Fermi level. It is interesting to find that upon their recombination, an intermediate akin to typical binuclear μ-η¹:η² nitrido-metal species in homogeneous complexes with comparable geometry, expected electron density and acceptable bond distances (we calculated an N-N distance of 1.16 Å cf. N−N distance in nitrido Zr complex was shown to be 1.196Å[65]) within experimental error to the two Ru atoms, is identified. Such surface complex may have gone through to the transition state of the linear μ-1,2 diazenido Ru−N≡N−Ru species since transition between diazenido-, hydrazido- and nitrido-metal has been reported in reversible N₂/H₂ activation in solution complexes[28]. This clearly suggests a similarity in N recombination chemistry as homogeneous counterparts with the important additional function of support in this single atom solid catalyst to remove the H in ammonia decomposition. Notice that the increasing interaxial overlap between 2p orbital electrons of two N at the expense of their overlap with 3d orbital electrons of Ru near the Fermi level indicating the multi bonding between two N but lesser bonding with Ru sites with expected bonding distances.

## Discussion

In this paper we demonstrate the strong metal support interaction of Ru on polar MgO(111) to suppresses hydrogen poisoning on Ru surface during ammonia decomposition hence enabling better heterolytic $NH_3$ activation and promoting the N−N recombination. Thus, the (111) facet of MgO is a more efficient surface to host Ru for ammonia decomposition, showing at least 4 times higher catalytic activity than those of (100) and (110) facets at a low temperature of 400 °C. Significantly, we have also found that this support surface can give atomically-dispersed Ru which displays superior performance in ammonia decomposition than Ru NPs.

Unexpectedly, the dispersed, isolated Ru atom at greater surface dilution does not demonstrate the highest specific activity per Ru as those typical single atom catalysts. The closed Ru pairs at optimal but moderate surface concentration without excessive coverage of the support exhibit even better specific activity. This finding enables us to confirm the cooperative single-atoms Ru solid catalysts in synergy in ammonia decomposition as akin to those of solution counterparts[28]. Based on our experimental and theoretical investigation, a new reaction mechanism is proposed. As the ammonia approaches the surface, it will be activated by the dispersed $[Ru^{2+}–O^{2-}]$ ion pairs at the metal support interface via heterolytic cleavage of N−H bonds. These attained protons are preferentially and rapidly hopping across the oxygen-terminated MgO(111) surface for H to transfer across the oxygen sites[65]. Consequently, the retained electrons enrich the electron density of Ru site, which enables electronic back donation to Ru−N antibonding orbital to weaken the strength of Ru−N bond for fast recombination to $N_2$ between the 2× surface Ru pairs. This mechanism clearly differs from the classical coordinated stepped Ru sites in cluster/nanoparticle and give hints to guide for better catalyst candidates with higher atom efficiencies. This class of attractive catalysts apparently achieves more superior catalytic performance in ammonia decomposition under low reaction temperatures. Importantly, our experimental and theoretical studies demonstrate a synergistic effect of the efficient hydrogen spillover and recombination of N atoms in closed Ru pairs with unique support interaction, which is beyond the capability of $B_5$ cluster sites. Our finding posts the importance of the utilization of synergistic single-atom noble catalysts in ammonia synthesis and decomposition, which can efficiently use the noble metals at the greatest extent.

## Methods

### Materials synthesis

**Preparation of different MgO supports.** MgO nanosheets with preferential exposed (111) facets, hereafter denoted as MgO(111), were prepared by using the hydrothermal method with the benzoic acid as the surfactant. In brief, 2.0 g $MgCl_2$ and 0.12 g benzoic acid were dissolved in 60 mL deionized water with sonication treatment at room temperature and then the obtained mixture was stirred for 10 min. 20 mL of 2 M NaOH solution was added drop by drop into the mixture with vigorous stirring, forming a white precipitate. The slurry was subsequently transferred to a 100 mL autoclave and gradually heated to 180 °C and maintained for 24 h. The obtained $Mg(OH)_2$ precursor was washed with deionized water, and dried at 80 °C under vacuum overnight after filtration. The MgO(111) nanosheets were obtained after calcination of the $Mg(OH)_2$ precursor in compressed air at 500 °C for 6 h[43].

MgO(110) with exposed (110) facets was synthesized by reconstruction of commercial MgO. The commercial MgO was purchased from Sigma Aldrich. Typically, 500 mg commercial MgO was boiled in water for 5 h with an oil bath, followed by drying at 120 °C overnight. The obtained precursor was calcined at 500 °C for 6 h under a vacuum system[45].

MgO(100) was obtained by traditional thermal decomposition of $Mg(NO_3)_2$ precursor directly at 500 °C for 5 h[46].

The MgO nano-cubes (denoted as MgO-cube) were prepared by combustion of Mg ribbons (99.98%, Sigma Aldrich) in an adequate oxygen gas flow. These smoke crystals were collected by a glass plate and rapidly transferred into sample tubes with Ar protection to prevent the surface etching in wet environments. The water solution with different values (pH = 2–6.8) was used to cleave the MgO-cube in (110) and (111) cuts of the ridges by dissolution of MgO surfaces. A well-control of surface cut with identified (111), (110) and (100) facets can be obtained by controlling the treatment time from 2 min to 7 days, finally transforming MgO nano-cubes into octahedron shape. The obtained precursors were carefully calcined under vacuum to remove the surface hydroxyl function groups to obtain clean MgO surfaces with different facets[55–57].

**Preparation of supported Ru catalysts.** Supported Ru catalysts with different MgO supports (~3 wt%) were synthesized by integrated methods including $Ru_3(CO)_{12}$ incorporation and decomposition of $Ru_3(CO)_{12}$. Briefly, $Ru_3(CO)_{12}$ was ultrasonic-dispersed in tetrahydrofuran (THF) solution to avoid the affection of water and then desired amount of MgO supports were added with vigorous stirring for 2 h at room temperature. After removing the solvent, the obtained orange powders were treated at 90 °C overnight under vacuum where pink or light grew powders were obtained (the color depends on the loading amounts of Ru and support choice). Then these precursors were heated and vaporized and deposited on MgO supports before reaching 300 °C with 2 °C/min for 3 h under Ar gas flow.

**Catalytic evaluation for ammonia decomposition.** The ammonia decomposition was carried out in a purposely built continuous flow fixed-bed reactor with a computer-controlled auto-sampling system. Typically, 50 mg of catalyst was loaded in the center of the quartz tube sandwiched with quartz wools. Prior to the reaction, the catalyst was pretreated at 350 °C under 5%$H_2$/He gas flow for 4 h. The gas was switched to $NH_3$ stream and then the catalyst bed was adjusted to the target reaction temperature. All the activity tests were conducted in the temperature range of 200-600 °C with varied weight hourly space velocity (WHSV) under atmosphere pressure. After the stabilization, the gas composition was analyzed by an online gas chromatography (Agilent 7890 A) equipped with TCD detector and a HayeSep Q column. Kinetic study was performed with varied temperatures and WHSV to ensure that the reaction conditions was within the kinetic zone. The Arrhenius plots and activation energies for ammonia decomposition were calculated based on the activity evaluated under the same gas flow and compositions but at different temperatures with the conversion kept below 15% so they were far from equilibrium. The activity was therefore at the kinetic controlled steady state. All the catalysts were evaluated twice to ensure the accuracy of the measurements. The reaction orders were conducted by varying one gas concentration of $N_2$, $H_2$ or $NH_3$ with He as the balance gas and measured at 350 °C.

It is noted that the contribution of each facet towards the ammonia decomposition rate was obtained independently: we first synthesized MgO cube with exclusive (100) facet; then we cleaved these cubes to expose (110) and (111) facets at various proportions (11 samples) and compared to the synthesized octahedron sample with exclusive (111) facet. Each sample was followed with the same Ru immobilization and tested for ammonia decomposition. Thus, the overall activity of each sample can be derived from the individual contributions of (100), (110), and (111) facets by resolving the equation as below.

$$\text{Conv.}(NH_3) = aX + bY + cZ$$

Whereas the a, b, c are the percentage of (100), (110), (111) facets in each sample; the X, Y, and Z are the individual conversion of pure (100), (110), or (111) facet.

By taken all 11 samples with comparable surface area into account, the activity for each facet can be derived, as shown in Fig. 2). As a result, the catalytic contribution of each facet can be obtained in the contour map.

**Characterizations.** XRD patterns were performed on a PANalytical X'Pert Pro diffractometer operating at 40 kV and 30 mA with a step size of 0.004°. Samples were dusted and pressed onto either a glass or an aluminum preparative slide.

TEM images were collected by using a Philips Analytical FEI Tecnai 30 electron microscope operated at an acceleration voltage of 300 kV. Fresh samples were dispersed ultrasonically, dropped and dried on copper grid with lacy support films.

Microscopy for Z-contrast STEM imaging and condition was performed on a JEOL ARM-200F Cold field emission source (CFEG), equipped with a Cs probe corrector to give a probe size of 0.8 Å, using 200 kV acceleration voltage and 22 to 25 mrad convergence angle. An annular dark field detector at 8 cm camera length was used for signal acquisition, with inner and outer collection angles of 72.80 and 235.75 mrad for HAADF imaging, respectively. All alignments and focusing were completed away from the areas imaged to reduce electron beam damage to the material under emission current 7μA. All micrographs were obtained without tilting the sample (alone zone axis [111] of MgO film). The HAADF-STEM mode uses an annular detector to capture the scattered electrons to high angles. Rutherford scattering from the atomic nuclei predominates of Ruthenium, with a cross-section proportional to $Z^2$. A high-Z number a Ruthenium atom on a low-Z support MgO therefore gives a spot with high contrast.

Non-linear filter[66] has been applied to increase the quality and contrast of the image, a low-pass filter and a Wiener filter across the Fourier transformation (FT) of the real-space image before reconstructing the image from the filtered FT. It can efficiently reduce noise without noticeable artifacts. The simulation of the elastic interaction of electrons applying the multislice method[67] to demonstrate HAADF-STEM imaging of MgO(111) catalyst, the contrast of Ru atoms and Mg atoms can be enhanced. Using supercells derived from DFT models (1x Ru on MgO(111), see Fig. S3; 2x Ru on MgO(111), see Fig. S17, etc), the implementation of Debye-Waller factor and absorption potentials were realised for each slice of the input structure. The input files for the model-based Rietveld refinement result are the unit cell to extract signal at periodic thickness levels up to maximum object thickness from 4 nm to 10 nm.

In-situ AP-XPS study were carried out by means of operando XPS measurements at the Diamond Light Source B07 beamline. The samples were dispersed in acetone and mounted on a silicon wafer, followed by heating at 110 °C to remove the solvent. The temperature was maintained at a constant throughout the testing at 300 °C. The measurements were performed either under 0.9 mbar Ar or 0.9 mbar $NH_3$. The gas composition was continuously monitored by online mass spectrometry (MS). If not otherwise stated, Ru 3d, O 1 s, N 1 s and Mg 2 s spectra were recorded using a fixed photon energy of 950 eV. The binding energy (BE) scale was calibrated with respect to the Mg 2 s (88.1 eV) as internal standard. Normal XPS measurement was conducted on an Omicron Sphera II photoelectron spectrometer equipped with Al-K$_\alpha$ X-ray radiation source hv = 1486.6 eV. The spectrometer is connected to a in situ chamber which can pretreat samples with different temperatures under different gases.

XAFS spectra was recorded at the Ru K absorption edge, in fluorescence mode using a Lytle fluorescence detector in B18, Diamond light source. A Si (111) Double Crystal Monochromator (DCM) was used to scan the photon energy. The energy resolution for the incident X-ray photons was estimated to be $2 \times 10^{-4}$. The Demeter

software package (Athena and Artemis) was used for XAFS data analysis for the Ru data. To ascertain the reproducibility of the experimental data, at least two scan sets were collected. The spectra were calibrated with foils as a reference. And the amplitude parameter was obtained from EXAFS data analysis of the Ru foil, which was used as a fixed input parameter in the data fitting to allow the refinement in the coordination number of the absorption element. In this work, the first shell data analyses under the assumption of single scattering were performed with the errors estimated by R-factor.

Temperature programmed surface reaction (TPSR) measurement of samples were carried out on a Micromeritics Autochem II ASAP 2920 hemisorption analyzer with a mass spectrometer detector. The activation procedure of catalysts was the same as that used for activity evaluation. After ammonia adsorption, the sample was heated from 50 to 800 °C with the purge of Ar. The signals of $H_2$ (m/z = 2), $NH_3$ (m/z = 17), and $N_2$ (m/z = 28) were tracked during the investigation. $H_2$-TPD was performed on the same Micromeritics Autochem II ASAP 2920 chemisorption analyzer and the signals of $H_2$ (m/z = 2) was monitored during the measurement.

The in-situ DRIFTS experiments for ammonia decomposition were carried out by a Thermo Scientific Nicolet 6700 spectrometer, equipped with ZnSe windows and a mercury cadmium telluride detector cooled by liquid nitrogen. Spectra were obtained by collecting 32 scans at a resolution of 4/cm and are presented in absorbance unit. Fresh samples were loaded onto the in-situ sample holder. The sample was then flushed with Ar for 30 min to clean the surface under required temperature. After collecting background spectra, $NH_3$ gas was passed through the sample holder for 30 min at 15 mL min$^{-1}$ and in-situ sample spectra were recorded after purging out the exceed ammonia gas by Ar. For the cycle testing, the DRIFTS for $NH_3$ transformation was collected under alternative sweeping between $NH_3$ and Ar at at 300 °C.

**Computational details.** Spin-polarized periodic density functional theory calculations were performed employing the Vienna Ab initio Simulation Package (VASP)[68]. While the interactions between the core electrons and nuclei were evaluated by the projector augmented wave (PAW)[69–71], the valence electrons were explicitly considered as follows: Mg(3s$^2$), O(2s$^2$, 2p$^4$), Ru(4d$^7$, 5s$^1$), N(2s$^2$, 2p$^3$) and H(1s$^1$). The generalized gradient approximation (GGA) within Perdew−Burke-Ernzerhof (PBE) functional[72] was applied to describe the exchange correlation of electrons. Furthermore, the van der Waals corrections with DFT-D3 scheme proposed by Grime et al. [73] was adopted to feature the interactions of atoms. The optimized structures were obtained by using a conjugated-gradient algorithm when the residue ionic forces are less than |0.01| eV/Å. A k-point grid of 1x1x1 and the plane wave basis set with a cutoff energy of 450 eV were used for all calculations. In addition, the maximally localized Wannier functions (MLWF) method using Wannier90 code has also been applied to plot the projected orbitals of considered atoms, see Fig. S13b[74].

MgO (111), (100) and (110) slab models with formula of Mg$_{100}$O$_{100}$, Mg$_{128}$O$_{128}$, and Mg$_{144}$O$_{144}$ were adopted to model the MgO surfaces exposed in the experiments. The MgO(111) slab has eight alternating anionic oxygen and cationic magnesium layers with the top being terminated by O atoms and the bottom by Mg atoms. To minimize the interactions between slabs, a vacuum space of 15 Å was generated for all slab models. The models were then optimized with their bottom two layers fixed in their bulk position, whilst the rest atoms and adsorbed species are allowed to relax. Also, the dipolar correction along z direction was included.

We compared the stability of an Ru atom adsorbed on the surface (adsorbed model) with that of an Ru atom substituted for a Mg atom (embedded model). The formation energies ($E_f$) of the MgO(111)-

adsorbed and embedded Ru models:

$$E_f(\text{adsorbed model}) = E(Ru(\text{adsorbed})/MgO(111)) - E(MgO(111)) - E(Ru)$$

$$E_f(\text{embedded model}) = E(Ru(\text{embedded})/MgO(111)) - E(MgO(111)) - E(Ru) + E(Mg)$$

where E(Ru(adsorbed)/MgO(111)), E(Ru(embedded)/MgO(111)), E(MgO(111)) are the total energies of the adsorbed and embedded Ru/MgO(111) models and the bare MgO(111) surface, while E(Ru) and E(Mg) are the atomic energies Ru and Mg in their pure hcp bulk structure.

Two Ru atoms (2Ru) supported on the MgO(111), (100) and (110) slab models, denoted 2Ru/MgO(111), 2Ru/MgO(100) and 2Ru/MgO(110) (Fig. S17), were used to represent the binuclear active site for ammonia decomposition. The adsorption energies of the intermediates of the stepwise dehydrogenation and N-N recombination from decomposition of two ammonia molecules on 2Ru/MgO models were calculated as follows:

$$E_{ads} = E(\text{Adsorbate}/2Ru/MgO) - E(2Ru/MgO) - 2*E(NH_3)_{(g)}$$

where E(Adsorbate/2Ru/MgO) and E(2Ru/MgO) are the total energies of the 2Ru/MgO using (111), (110) or (100) facet with adsorbate and in the bare form, respectively. Adsorbate refers to all $NH_3$, $NH_2$, NH, N and $N_2$ molecules. $E(NH_3)_{(g)}$ refers to the total energy of a single $NH_3$ molecule in gas phase. The adsorption energies of intermediates of the dehydrogenation of NH* and H spillover was also calculated on atomic Ru supported on MgO(111), (110) and (100) as follows:

$$E_{ads} = E(\text{Intermediate}/Ru/MgO) - E(Ru/MgO) - 0.5*E(N_2)_{(g)} - 0.5*E(H_2)_{(g)}$$

where E(Intermediate/Ru/MgO) and E(Ru/MgO) are the total energies of the Ru/MgO using (111), (110) or (100) facet with adsorbed NH* or N* + H* and in the bare form, respectively. $E(N_2)_{(g)}$ and $E(H_2)_{(g)}$ are the total energies of $N_2$ and $H_2$ gas molecules. Spillover barriers were calculated by using climbing-image nudged elastic band (CI-NEB) approach[75]. To obtain the transition state structures, six images were created in each path and optimized with force threshold of |0.02| eV/Å. The atomic charges were calculated by applying the Bader method suggested by Henkelman et al[76–79].

## Data availability

All the data in this study are available in the manuscript or Supplementary Information. Correspondence and requests for materials should be addressed to the corresponding author.

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

## Acknowledgements

The financial support of this work from the EPSRC research council of UK and Siemens, plc are acknowledged. H.F. acknowledges the receipt of a scholarship from China Scholarship Committee (CSC) to Oxford University, UK. S.W. thanks to Siemens, plc and EPSRC for a joint DPhil Studentship. The authors also acknowledge the use of STEM facilities of National Institute of Advanced Industrial Science and Technology of Japan; the use of AP-XPS facilities in the beamline B07 and XAS from beamline B18 of Diamond Light Source, UK. H.V.T. thank to the support of NAFOSTED as a statement "This research is funded by Vietnam National Foundation for Science and Technology Development (NAFOSTED) under grant number 104.06-2020.50 (H.V.T.)".

## Author contributions

H.F. and S.W. carried out material synthesis, laboratory characterization, and materials evaluation with the assistance from T.A., J.Z., K.C.L. and J.F. (S)TEM and EELS were collected and analyzed by P.-L.H., R.K. and K.S. XAS analysis by S.W. AP-XPS were collected by H.F., S.W., A.L. and G.H. Y.I.A.R., H.Y.T.C. and H.V.T. performed the DFT simulations and interpretation. The manuscript was written and revised by H.F. and S.C.E.T. The project was planned and directed by S.C.E.T. All authors discussed the results and commented on the manuscript.

## Competing interests

The authors declare no competing interests.
