## [Peer Review File · Nature Communications]

Reviewers' comments:

Reviewer #1 (Remarks to the Author):

This manuscript describes the preparation, characterization, and modeling of MgO-supported Ru catalysts for NH₃ decomposition. NH₃ decomposition catalysis is practically relevant to use of NH₃ as a clean hydrogen storage material. The work is thoroughly reported and experimental observations likely of interest to the community. The inferences regarding the nature of active sites drawn from these observations would benefit from more careful, critical analysis. Further, the connections between experimental observation and computational models is weak. Following are comments for author consideration in revision:

1. Authors might cite doi:10.1016/j.jcat.2004.12.013 with reference to the connections between ammonia synthesis and decomposition.
2. The primary observation of the manuscript is that NH₃ decomposition rates, normalized to mass Ru, are greater on supports created to preferentially expose the (111) facets of MgO (Figure 2). Authors use these observations to argue that Ru atomically dispersed on this polar facet exhibit high intrinsic (site-normalized) rates. The difficulty in drawing this conclusion is that mass normalization does not necessarily report on available active sites. Have authors considered some approach to titrate or quantify active sites, or even just surface-exposed Ru?
3. Are experimental observations robust to time-on-stream? For instance, Fig 2a, do conversion profiles follow the same path during T-ramp down?
4. Figure 2d, are rates measured at identical gas compositions? Corrected for approach to equilibrium?
5. Figure 2e,f, what is the mechanistic rationalization for observed differences in rate orders? Authors make reference to role of (111) facet in H₂ desorption. A schematic to illustrate the hypothesis would make the arguments and relationship to experimental observations (eg Fig 3) easier to follow.
6. Assertions on lines 372-377 need references and/or stronger justification.
7. Figure 4, differences in mass-normalized rate with loading are approximately a factor of two. What does this mean with respect to the statement on lines 414-416? Again, without some ability to relate rates to numbers of sites, conclusions regarding the relative effectiveness of sites seems tenuous.
8. The connections between the DFT model and experimental observations are at best weak. Authors consider only one model of an "active site," neglect a key element of the posited mechanism ("protons on MgO left the surface to form H₂"), and do not make quantitative (or even qualitative) connections with observed kinetics. One even wonders about the robustness of the model; MgO(111) is polar, and thus results will be highly sensitive to slab termination. The fact that Bader charges on a nominally Ru(O) atom are +1.5 suggests charge transfer to the opposite side of the slab that is not physical. Without

more careful justification, testing, and quantification of the DFT results, they add little to the overall manuscript.

Reviewer #2 (Remarks to the Author):

In this manuscript the authors test, characterize and model several Ru/MgO catalysts for the decomposition of ammonia. Different Ru loadings and MgO facets are studied and DFT calculations are performed to infer the mechanism of the reaction. The article concludes that binuclear Ru pairs and Ru(2+)-O ensembles are the most active for the reaction.

The manuscript is well written and the data is well presented, however I do not recommend publication of the manuscript at this stage in Nat Commun for the following reasons:

1. MgO sheets exhibiting {111} facets are prepared, but the paper shows these sheets to contain, in addition to {111} facets, {110} and {100} facets as well. Also, it is claimed that Ru atoms are present on all these three surfaces. Then the question is, how do the authors know the catalytic contribution of each facet? In general the manuscript lacks from detailed quantification of the facets exposed by MgO, not only this applies to MgO sheets, but also to all the other materials prepared. It is very important to quantify the facets present. This can be done, for instance, by IR analysis at low temperature of probe molecules. The manuscript would be strongly reinforced by providing accurate quantification data regarding the facets exposed.
2. For an unambiguous identification of Ru atoms in the STEM images, authors should provide a direct proof of the identification of Ru atoms. Atomic-resolved EELS or EDX could work.
3. NAPP-XPS data is apparently in contradiction to the mechanism proposed. By NAPP-XPS it is concluded that Ru is found as +2 species under Ar stream, and then it shifts to Ru(0) during ammonia decomposition. However, the authors claim that a heterolytic cleavage between Ru(2+)-O and N-H occurs during the reaction, which results first in electron donation from N to Ru, and then electron back-donation to Ru-N antibonding orbital. Is this the case, then why Ru appears reduced in the NAPP-XPS experiments? If the answer is the use of low pressure in the NAPP-XPS experiments resulting in quenching of species, then the authors should provide a complete set of experiments at different pressure values.
4. The data compiled in Table 1 is rather incomplete. Other supported Ru catalysts are known for the decomposition of ammonia which are not included in the table.
5. Finally, I'm surprised by the claim of binuclear Ru pairs. I do not see them in Figure 4. The catalysts labeled (4) and (5), which are the most active, contain a plethora of Ru ensembles. In fact, it is hard to recognize binuclear associations at all...

Reviewers' comments:

Reviewer #1 (Remarks to the Author):

This manuscript describes the preparation, characterization, and modeling of MgO-supported Ru catalysts for NH₃ decomposition. NH₃ decomposition catalysis is practically relevant to use of NH₃ as a clean hydrogen storage material. The work is thoroughly reported and experimental observations likely of interest to the community. The inferences regarding the nature of active sites drawn from these observations would benefit from more careful, critical analysis. Further, the connections between experimental observation and computational models is weak. Following are comments for author consideration in revision:

Reply: Thank you very much for your constructive comments. These comments are all valuable and helpful for improving our article. We have added more data and discussions in the revised manuscript for more careful and critical analysis. In addition, we have put some great efforts on connecting the relationship between the experimental study and DFT calculations, which could be useful for readers to get insights into the nature of catalysis in our case.

1. Authors might cite doi:10.1016/j.jcat.2004.12.013 with reference to the connections between ammonia synthesis and decomposition.

Reply: Thank you very much for your suggestion. We have added the reference in the manuscript.

[7] A. Boisen, S. Dahl, J. K. Nørskov, C. H. Christensen, Why the optimal ammonia synthesis catalyst is not the optimal ammonia decomposition catalyst. *J. Catal.* 230, 309–312 (2005).

2. The primary observation of the manuscript is that NH₃ decomposition rates, normalized to mass Ru, are greater on supports created to preferentially expose the (111) facets of MgO (Figure 2). Authors use these observations to argue that Ru atomically dispersed on this polar facet exhibit high intrinsic (site-normalized) rates. The difficulty in drawing this conclusion is that mass normalization does not necessarily report on available active sites. Have authors considered some approach to titrate or quantify active sites, or even just surface-exposed Ru?

Reply: Thank you very much for his/her nice suggestion. We have now included the

CO titration over the different samples to quantify the available active metal sites on the surface of catalysts. The turnover frequency (TOF) values were based on the CO chemisorption value, indicating the mole of ammonia converted by per CO site at the catalyst surface per second ($\text{mol-ammonia mol-site}_{\text{CO}}^{-1} \text{ s}^{-1}$, for short, s^{-1}). To ensure getting a reliable initial rate, the ammonia conversion for TOF calculation was maintained at lower than 15%. As shown in **Table S7**, the Ru/MgO(111) exhibits a TOF values of 1.67 s^{-1} , much higher than those of Ru/MgO(110) and Ru/MgO(100). This result again indicates the higher Ru dispersity on MgO(111) facet coupled with higher activity per site for ammonia decomposition, which shows a similar trend with those result normalized to mass Ru. We have added related experiments and discussion in the revised manuscript.

Table S7 Rates of ammonia decomposition over the MgO supported Ru catalysts.

Sample	Ru content / wt.% ^a	CO uptake / $\mu\text{mol g}_{\text{cat}}^{-1}$	Dispersion / %	Activity / $\mu\text{mol NH}_3 \text{ g}_{\text{Ru}}^{-1} \text{ s}^{-1}$	TOF / $\text{s}^{-1\text{b}}$
Ru/MgO(111)	0.102	0.68	67.4	11160	1.67
Ru/MgO(110)	0.113	0.58	51.9	1850	0.36
Ru/MgO(100)	0.108	0.49	45.9	1290	0.28

^a Obtained by ICP-OES analysis; ^b TOF values are based on the CO chemisorption uptake over different samples with <15% conversion; reaction conditions: T=400 °C, WHSV = 30,000 mL $\text{g}_{\text{cat}}^{-1} \text{ h}^{-1}$, 1 bar.

3. Are experimental observations robust to time-on-stream? For instance, Fig 2a, do conversion profiles follow the same path during T-ramp down?

Reply: Thank you for your questions and comments. In our cases, the Ru/MgO catalysts are stable during the ammonia decomposition. Typically, the catalytic performance and stability of Ru/MgO(111) for ammonia decomposition under different temperatures was evaluated, as shown in **Fig. S19**. The catalyst was first tested at 425 °C and then the temperature was ramped down step by step. Each step was held on for 20 h on stream. According to the **Fig. S19**, the Ru/MgO(111) displays a robustness in activity under different temperatures during T-ramp down and no obvious deactivation was observed in our case. It is worth noting that the conversion of ammonia decomposition under different temperatures is similar with the catalytic activity shown in Fig. 2a during T-ramp up. Furthermore, when the temperature was raised up to 400 °C, the ammonia conversion is 84.7%, which is almost the same with the initial activity

obtained, further confirming the robustness in activity of the Ru/MgO(111). We have added related experiments and discussion in the revised manuscript.

Fig. S19. The catalytic performance of ammonia decomposition and stability of Ru/MgO(111) as a function of step-by-step temperature ramp down and up.

4. Figure 2d, are rates measured at identical gas compositions? Corrected for approach to equilibrium?

Reply: Thank you for raising this point. In our case, the Arrhenius plots and activation energies for ammonia decomposition were calculated based on the activity evaluated under the same gas flow and compositions but at different temperatures with the conversion kept below 15% so they were far from equilibrium. The activity was therefore at the kinetic controlled steady state. All the catalysts were evaluated twice to ensure the accuracy of the measurements. We have added more descriptions in the revised manuscript for better understanding.

5. Figure 2e,f, what is the mechanistic rationalization for observed differences in rate orders? Authors make reference to role of (111) facet in H₂ desorption. A schematic to illustrate the hypothesis would make the arguments and relationship to experimental observations (e.g. Fig 3) easier to follow.

Response: Thank you for the constructive suggestion. The difference in reaction order between different MgO supported Ru catalysts is due to the different adsorption extent towards hydrogen on the catalyst surface, which results in the difference in their reaction rate. Based on our experiments and analysis, the typical Ru/MgO(100) and Ru/MgO(110) show large and negative hydrogen order (γ), -0.85 and -0.79, respectively. This is due to the strong adsorbed H atoms on the catalyst surface, well-known as hydrogen poisoning. In contrast, the exposed MgO(111) surface exhibits a strong affinity for hydrogen activation (via Frustrated Lewis pair with Ru) and removal

(proton hopping), which helps to reduce the hydrogen poisoning. In addition, the reduction of Ru during hydrogen activation can also promote the N-N recombination by donating the electrons to the Ru-N antibonding orbitals, corresponding to the large α value. To facilitate a better understanding to readers, a schematic is incorporated to explain the difference of reaction orders between different samples, which illustrates the reaction pathways. We have added related descriptions in the revised manuscript.

Small ammonia order (α) and large hydrogen order (γ) indicate that N species such as N and NH groups, and H adatoms are strongly adsorbed onto the catalyst surface.

Scheme S1 The schematics demonstrating the difference between the Ru/MgO(111) and Ru/MgO(100).

6. Assertions on lines 372-377 need references and/or stronger justification.

Response: Thank you for your excellent suggestion. R. Prins et al. (J. Phys. Chem. C 2012, 116, 14274–14283) studied the hydrogen spillover on the nonreducible MgO support. They reported that hydrogen spillover to defect surfaces of MgO(100) is difficult while it is possible when the surface possesses defects to expose more oxygen-containing groups. In our previous work (J. Am. Chem. Soc. 2021, 143, 9105–9112; ACS Catal. 2020, 10, 5614–5622), we also found that the MgO(111) displayed an excellent hydrogen spillover phenomenon to reduce the hydrogen poisoning on the Ru surface. Further combinations of in situ spectroscopic and STEM imaging technique help us to derive the atomic pathway for reversible hydrogen migration on a nonreducible metal oxide in this manuscript that we observed the hydrogen hopping to MgO(111) surface via H₂-TPD and DRIFTS experiments. For better understanding, we have added related references in the revised manuscript.

7. Figure 4, differences in mass-normalized rate with loading are approximately a factor of two. What does this mean with respect to the statement on lines 414-416? Again, without some ability to relate rates to numbers of sites, conclusions regarding the relative effectiveness of sites seems tenuous.

Reply: Thank you for the valuable suggestion. To avoid the controversy of using the mass-normalized rate as y-axis, we have now carried out CO chemisorption on exposed

Ru sites and derived their TOF values, as shown in Table S8. Fig. 4 shows the TOF values as a function of Ru loading from extremely low 0.01 wt.% to 7.0 wt.%, it can be seen that it also gives a double volcano-type curve with the increase Ru loading. When the Ru surface concentration on MgO(111) becomes higher such as 2×Ru and 3×Ru atom ensembles are expected to be higher as the Ru loading increases. Interestingly, the TOF value increases from 1.67 to 3.43 s⁻¹ first and then decreases at the increase of Ru loading. This reflects to the synergy on neighbor surface Ru atoms on the MgO(111) for the catalytic decomposition. We have replaced the mass-normalized rates to the metal site-normalized rates in Fig.4 and added more discussions in the revised manuscript.

Here, we want to clarify that the anticipated mechanism requires 2 neighbor Ru sites for the decomposition and formation of N₂ instead of a single Ru atom (see DFT results). However, we do not imply the active site to a binuclear Ru of fixed geometry but rather a plethora of Ru ensembles (with at least two neighbor Ru atoms for the catalysis). As a result, we have changed the title from "Dispersed Binuclear Ru Pairs for Catalytic Ammonia Decomposition" to "Dispersed Surface Ru Ensembles on MgO(111) for Catalytic Ammonia Decomposition" to avoid the misunderstanding.

Fig. 4 (a) The TOF values for ammonia decomposition as a function of Ru loading at 400 °C; (b) Typical STEM images of Ru/MgO(111) with different loadings, the labeled

numbers are corresponding to the samples in Fig. 4(a); the scale bar is 1 nm.

Table S8 Rates of ammonia decomposition over the MgO supported Ru catalysts.

Sample	Ru content ^a / wt.‰	CO uptake / $\mu\text{mol g}_{\text{cat}}^{-1}$	Dispersion / %	Activity / μmol $\text{NH}_3 \text{g}_{\text{Ru}}^{-1} \text{s}^{-1}$	TOF ^b / s^{-1}
	0.102	0.68	67.4	11160	1.67
	0.197	1.30	66.5	12091	1.84
	0.309	2.02	66.0	14881	2.28
	0.388	2.51	65.5	18601	2.87
	0.786	4.84	62.3	21159	3.43
	1.05	6.35	61.1	19520	3.23
	1.91	11.26	59.6	15625	2.65
	2.95	17.13	58.7	13481	2.32
	5.1	29.06	57.6	12887	2.26
Ru/MgO(111)	9.4	51.60	55.5	12332	2.25
	20.7	105.65	51.6	11573	2.27
	31.2	150.91	48.9	11281	2.33
	35.5	174.52	49.7	12755	2.59
	41.1	193.10	47.5	13765	2.93
	45.4	210.61	46.9	13228	2.85
	54.3	241.69	45.0	12301	2.76
	61.2	266.96	44.1	10624	2.44
	70.9	306.46	43.7	10204	2.36
	104	398.10	38.7	8229	2.15

^aObtained by ICP-OES analysis; ^b TOF values are based on the CO chemisorption over different samples with <15% conversion; reaction conditions: T=400 °C, WHSV = 30,000 mL $\text{g}_{\text{cat}}^{-1} \text{h}^{-1}$, 1 bar.

8. (A) The connections between the DFT model and experimental observations are at best weak. (B) Authors consider only one model of an "active site," neglect a key element of the posited mechanism ("protons on MgO left the surface to form H₂"), and do not make quantitative (or even qualitative) connections with observed kinetics. (C) One even wonders about the robustness of the model; MgO(111) is polar, and thus results will be highly sensitive to slab termination. (D) The fact that Bader charges on a nominally Ru(0) atom are +1.5 suggests charge transfer to the opposite side of the

slab that is not physical. Without more careful justification, testing, and quantification of the DFT results, they add little to the overall manuscript.

Reply: We agree and thank the Reviewer for the critical comment. Here is our point-by-point response to the queries about the DFT calculations.

(A) With regards to the weak connection of the DFT data with the experiment, we have revised our DFT calculations analysis to be more aligned with our experiments. For instance, the kinetic experiments reveal that the exceptional NH_3 decomposition activity of Ru/MgO(111) is associated with more positive order of reaction with respect to hydrogen. The kinetic experiments imply that using the (111) facet reduces the hydrogen poisoning of catalytic Ru surface. To confirm this, we analyzed the case of the removal of H from an $^*\text{NH}$ species adsorbed on a Ru single atom supported on MgO(100), MgO(110) and MgO(111), see Fig. 6 below. On Ru/MgO(100) and Ru/MgO(110), H atoms removed from the $^*\text{NH}$ species, remain on the Ru surface as their migration to the support surface is not energetically favorable. In contrast, the $^*\text{NH}$ on Ru/MgO(111) interacts with the surface O atoms of MgO in the periphery of Ru. This leads to the cleavage of the N-H bond wherein the H atoms move directly to the surface O atoms of the support. Thus, on Ru/MgO(111), the reduced H poisoning observed in the kinetic experiments is explained by the favorable spillover of H to the MgO surface. Furthermore, analysis of the Bader charges shows that the spillover H donates electron density to Ru, consistent with the simultaneous reduction of Ru and increase of surface OH⁻ species observed in the AP-XPS analysis. We further compared the complete NH_3 decomposition reaction coordinate using models of $(\text{NH}_3\text{-Ru})_2$ on MgO (100), (110) and (111). On $\text{Ru}_2/\text{MgO}(100)$ and $\text{Ru}_2/\text{MgO}(110)$, initial H removal from adsorbed NH_3 species are unfavorable, because the H atoms, unable to spillover to MgO, crowding to the atomically dispersed Ru catalyst. In contrast, the adsorption of NH_3 on $\text{Ru}_2/\text{MgO}(111)$ immediately leads to removal of H atoms and their migration to the MgO surface. Subsequent removal of H atoms from NH_x species are also thermodynamically favorable on $\text{Ru}_2/\text{MgO}(111)$ due to absence of crowding of the Ru catalyst and abundance of surface O atoms where H can migrate. The detailed revision of the discussion in the manuscript are shown below.

(B) With regards to the modeling of active sites, the revised DFT analysis now include models of Ru_1 and Ru_2 active sites on supported on MgO (100), (110) and (111) facets with complete elucidation N_2 formation from two molecules of NH_3 . Our results and conclusions are consistently coherent across these models and in agreement with the experimental data.

We focused on the effect of the catalyst structure on elementary steps leading to N-N recombination, the rate-determining step (*J. Phys. Chem.* **1987**, *91* (20), 5302–5307.) It is believed that spillover H will go back to Ru sites to acquire electrons and produce H_2 but this process is fast and does not affect the experimentally observed ammonia decomposition activity. Thus, to avoid distraction, we maintain the exclusion of the H_2 desorption step in the DFT simulations. Instead, we chose to account for the effect of

the unremoved of H* atoms on Ru for Ru₂/MgO(100) and Ru₂/MgO(110) and spillover H-O on Ru₂/MgO(111) on the thermodynamics of the steps toward N-N recombination.

(C) With regards to modeling of MgO(111), we agree that modeling polar surfaces is not trivial and careful testing is necessary. In our case, models of MgO(111) has been well studied.^(cited) Specifically, our group has tested various models of MgO(111) and Ru/MgO(111) before (*J. Phys. Condens. Matter* **2022**, *34*, 214007.) and have shown that the model like the one used in the current work provides consistent structural, energetic and electronic properties that agree with previous calculations and experiments. Thus, on top of the very good agreement of our DFT data with the current experimental results, we believe previous tests and DFT modeling of MgO(111) sufficiently validates the robustness of models and results.

(D) With regards to the question of charge transfer, we clarify that the Ru does not transfer charge to the opposite side of the slab. Instead, the electron density from Ru is donated to and shared by the surface O atoms in the periphery of Ru (Fig. S16b). This result agrees well with our previous results (*ACS Catal.* **2020**, *10* (10), 5614–5622; *J. Am. Chem. Soc.* **2021**, *143* (24), 9105–9112) and our current experimental XPS results.

Fig. S16b Bader charge of Ru (pink) and the change in Bader charge upon binding of Ru compared to bare MgO(111) of surface O atoms in the periphery of bound Ru (blue and cyan).

Fig. 6 Energy profile of H removal from NH and spillover to MgO on Ru₁/MgO(100), Ru₁/MgO(110) and Ru₁/MgO(111) shown in blue, green and red lines, respectively. The optimized structure of each intermediate is depicted wherein Mg, O, Ru, N and H atoms are depicted as orange, red, purple, blue and gray spheres; the values near the atoms of H, N and Ru are their respective Bader charges (in |e|). The value near each intermediate refer to the adsorption energy (in eV) with respect to the energies of Ru/MgO surface, a single H₂ and N₂ gas. E_a (in eV) refers to the activation energies of hydrogen spillover from the NH to the three MgO supports. The values in the inserted structure

Fig. 7 Reaction coordinate diagram of N₂ formation from two molecules of NH₃ on two Ru single atom catalysts supported on MgO(100), MgO(110) and MgO(111), shown in blue, green and red lines, respectively. The optimized structure of each intermediate is depicted wherein Mg, O, Ru, N and H atoms are depicted as orange, red, purple, blue and gray spheres.

Reviewer #2 (Remarks to the Author):

In this manuscript the authors test, characterize and model several Ru/MgO catalysts for the decomposition of ammonia. Different Ru loadings and MgO facets are studied and DFT calculations are performed to infer the mechanism of the reaction. The article concludes that binuclear Ru pairs and Ru(2+)-O ensembles are the most active for the reaction.

We thank the Reviewer's comments and suggestions. As per the suggestion, we have added much more data and discussions in the revised manuscript.

The manuscript is well written and the data is well presented, however I do not recommend publication of the manuscript at this stage in Nat Commun for the following reasons:

1. MgO sheets exhibiting {111} facets are prepared, but the paper shows these sheets to contain, in addition to {111} facets, {110} and {100} facets as well. Also, it is

claimed that Ru atoms are present on all these three surfaces. Then the question is, how do the authors know the catalytic contribution of each facet? In general, the manuscript lacks from detailed quantification of the facets exposed by MgO, not only this applies to MgO sheets, but also to all the other materials prepared. It is very important to quantify the facets present. This can be done, for instance, by IR analysis at low temperature of probe molecules. The manuscript would be strongly reinforced by providing accurate quantification data regarding the facets exposed.

Reply: Thank you for your comments and constructive suggestions.

We agree with the reviewer that a more detail quantification of the facets is required. To get a stronger quantitative correlation regarding the facet exposure as according to the reviewer's suggestion, we have now carried out the Diffuse reflectance infrared spectroscopy (DRIFTS) experiments at low temperature by using CO₂ as a probe molecule. Note that CO₂ as an acidic molecule can be adsorbed on different MgO surfaces with different basicities. The DRIFTS was conducted by a Thermo Nicolet 6700 IR spectrometer fitted with a liquid nitrogen cooled detector, DRIFTS accessory with an in-situ sample cell. All the MgO samples were heated to 400 °C for 15 min under Ar gas flow to clean the surface and then cooled down to 10 °C with condensate circulating water. After collecting background spectra, the sample is first contacted with CO₂ for 30 min for sufficient adsorption and in-situ sample spectra were recorded after purging out the exceed CO₂ gas by Ar.

Fig. S20 shows the recorded DRIFTS of adsorbed species over different MgO samples after CO₂ adsorption under Ar gas flow. It can be seen that the signals of adsorbed species over MgO(111), MgO(110) and MgO(100) are totally different, which is attributed to the different interaction between exposed facets and CO₂ probe. To further identify these species, Table S5 summarizes the IR results from the experimental studies, where the vibrational frequency bands recorded on MgO surfaces have been assigned either to monodentate, bidentate and tridentate. In our work, no hydrogen carbonate (1220 cm⁻¹) can be found on the surface. Compared with our recorded spectra, the MgO(100) shows the typical peaks at 1747 and 1326 cm⁻¹ assigned the vibration of tridentate, which is due to the exposed MgO(100) facet with interval Mg and O arrangement. The MgO(110) displays quite different peaks from Fig. S20 The peaks at 1710 and 1637 cm⁻¹ are ascribed to the typical vibration of bidentate, contributed by exposed MgO(110) facets. It is worth noting that the MgO (111) sample exhibits more different peaks. Table S6 shows the identified species towards exposed facets and summaries the results of the facet percentages over these MgO samples via spectral

deconvolution and integration. The MgO(100) exposes almost (100) facet while the MgO(110) possesses both (110) and (111) facets with 86.1% and 13.9%. It can be seen that the MgO(111) sample presents 66.4% of (111) facet, 23.5% (110) facet and 10.1% (100) facet, respectively. To show the quantification of the facets present, we have incorporated the related data and discussion in the revised manuscript.

Fig. S20 Recorded DRIFTS of adsorbed species over MgO(111), MgO(110) and MgO(100) after CO₂ adsorption under Ar gas flow.

Table S5 Summary of literature experimental attributions of IR bands assigned to carbonates species adsorbed on MgO surface.

Species	Structure	Carbonates			Ref.
		ν_{3high}	ν_{3low}	ν_1	
monodentate		1510-1550	1390-		21,22
		1550	1410	1035-1050	
		1520	1410	1050	
		1590,	1730	1060	
		1510	1415,		
		1450			

Bidentate		1385, 1335 1659,1626 1665-1710 1670, 1630	1273 1325- 1330 1270 1315, 1280	1024, 947 1005-1030	21,22
		1670, 1630	1270 1315, 1280	1000,850,950	
Tridentate		1325-1330, 1745-1750			21,22

Ref:

21. Y Cornu, D.; Guesmi, H.; Krafft, J.M.; Lauron-Pernot, H. Lewis Acido-Basic Interactions between CO₂ and MgO Surface: DFT and DRIFT Approaches. *J. Phys. Chem. C* **116**, 6645–6654 (2012).

22. Y Mutch, G.A.; Shulda, S.; McCue, A.J.; Menart, M.J.; Ciobanu, C.V.; Ngo, C.; Anderson, J.A.; Richards, R.M.; Vega-Maza, D.; *J. Am. Chem. Soc.* **140**, 4736–4742 (2018).

Table S6 Summary of the percentage of exposed facets over different MgO samples based on low temperature CO₂ adsorption.

Sample	Peak / cm ⁻¹	Facet	Area / %	Facets percentage / %		
				(100)	(110)	(111)
MgO(100)	1747	100	33.9			
	1670	100	1.4	98.6	1.4	0.0
	1326	100	64.7			
MgO(110)	1710	110	43.7			
	1637	110	36.8			
	1410	111	5.3	0.0	86.1	13.9
	1362	110	5.6			
	1270	111	8.6			
	1640	110	23.5			
MgO(111)	1528	111	4.2			
	1459	111	4.7			
	1425	111	6.9			
	1400	111	3.0	10.1	23.5	66.4
	1370	111	30.3			
	1330	100	10.1			
	1270	111	17.3			

As a result, we have cross-checked the total activity of the Ru/MgO (111) nanosheet sample which contains (111), (110), and (100) facets in the above percentage from CO₂ adsorption by taken the activity of each (100), (110), and (111) facets derived from contour map (see Table S9). We have found an excellent agreement showing the acceptable quantitative analysis of the facet activity within experimental errors.

It is noted that the contribution of each facet towards the ammonia decomposition rate was obtained independently: we first synthesized MgO cube with exclusive (100) facet; then we cleaved these cubes to expose (110) and (111) facets at various proportions (11 samples) and compared to the synthesized octahedron sample with exclusive (111) facet. Each sample was followed with the same Ru immobilization and tested for ammonia decomposition. Thus, the overall activity of each sample can be derived from the individual contributions of (100), (110), and (111) facets by resolving the equation as below.

$$\text{Conv. (NH}_3\text{)} = aX + bY + cZ$$

Whereas the a, b, c are the percentage of (100), (110), (111) facets in each sample; the X, Y, and Z are the individual conversion of pure (100), (110), or (111) facet.

By taken all 11 samples with comparable surface area into account, the activity for each facet can be derived, as shown in Fig. 2). As a result, the catalytic contribution of each facet can be obtained in the contour map.

2. For an unambiguous identification of Ru atoms in the STEM images, authors should provide a direct proof of the identification of Ru atoms. Atomic-resolved EELS or EDX could work.

Reply: Thank you for suggestion. In response to the comment, atomic-resolved EELS was carried out and the result is shown in Fig. S3.

High angle annular dark field (HAADF) imaging by 60 kV STEM coupled with electron energy loss spectroscopy (EELS) was performed to examine the precise position of individual Ru “atomic” sites on the oxygen surface of MgO(111). HAADF-STEM imaging revealed sites scattered across the surface that have a higher contrast than the surrounding top Mg sites (Figure 1d). The Z-contrast nature of HAADF imaging implied that this is due to a heavier constituent atom in the measured spectra. Simultaneous atomic EELS (Fig. S3b) was performed and the spectra confirm the presence of a Ru atom in the atomic column with the fingerprint Ru M_{2,3,4,5} edges. We have added the related experiments and description in the revised manuscript.

Fig. S3 (a) HAADF-STEM image of Ru/MgO(111) and (b) simultaneous acquisition EELS extracted on oxygen atom (green circle) and Ru atom (Orange circle) in **Fig. S3a**.

3. NAPP-XPS data is apparently in contradiction to the mechanism proposed. By NAPP-XPS is concluded that Ru is found as +2 species under Ar stream, and then it shifts to Ru(0) during ammonia decomposition. However, the authors claim that a heterolytic cleavage between Ru(2+)-O and N-H occurs during the reaction, which results first in electron donation from N to Ru, and then electron back-donation to Ru-N antibonding orbital. Is this the case, then why Ru appears reduced in the NAPP-XPS experiments? If the answer is the use of low pressure in the NAPP-XPS experiments resulting in quenching of species, then the authors should provide a complete set of experiments at different pressure values.

Reply: Thank you for the comment. We believe that there is a misunderstanding probably we did not give a clearer explanation. The Ru in Ru/MgO(111) is found as +2 species under Ar stream (non-reducing conditions) due to the charge transfer of Ru to nearby oxygen species on MgO(111) support (agreed with the DFT calculations) whereas Ru⁰ species is found in Ru/MgO(100) under identical conditions, as shown in **Fig. 3f and Fig. S13**. The positive charge of Ru species in Ru/MgO(111) is due to the adaptation of the stable configuration of Ru²⁺ with terminal O²⁻ (charge transfer) in MgO(111) surface. In the process of ammonia decomposition over the Ru/MgO(111), the cleavage of N-H bonds would occur at the interface, and the O terminations of MgO(111) only accept protons and allow their hopping. Therefore, the retained electrons of the N species enrich the electron density of Ru, resulting in the shift to higher binding energy of Ru species under ammonia atmosphere. In addition, the strength of Ru-N bond is highly dependent on the electronic state of Ru species. The enrichment of electrons during the cleavage of N-H bonds is beneficial for weakening the Ru-N bonds

via back donation of electrons to Ru–N antibonding orbitals, reducing the N retardation and promoting the rate of N–N recombination. For better understanding, we have revised our description in the revised manuscript.

4. The data compiled in Table 1 is rather incomplete. Other supported Ru catalysts are known for the decomposition of ammonia which are not included in the table.

Reply: Thank you for the useful suggestion. In light of the comment, we have now added further representative reports from literature in Table 1 in the revised manuscript.

Table 1 NH₃ conversion over Ru catalysts under atmospheric pressure.

Catalyst	Ru / wt%	T / °C	WHSV / mL g _{cat} ⁻¹ h ⁻¹	Conv. / %	H ₂ formation rate / mmol g _{Ru} ⁻¹ min ⁻¹	Ref.
Ru/Al ₂ O ₃	10	450	30,000	31.5	115	43
Ru/SiO ₂	10	450	30,000	34.5	114	43
Ru/MCM-41	5	450	30,000	42.4	284	44
Ru@ZrO ₂	3	450	30,000	ca. 40	447	45
Ru/CNTs	5	450	30,000	43.7	292	21
Ru/MgO	4.8	450	30,000	30.8	215	46
Ru/TiO ₂	4.8	450	30,000	27.2	190	46
Ru/Al ₂ O ₃	4.8	450	30,000	23.3	163	46
Ru/AC	4.8	450	30,000	28.7	200	46
Ru/CNFs	3.2	500	6,500	99.0	206	47
Ru/Cr ₂ O ₃	5.0	600	30,000	99.9	667.5	48
Ru/La _{0.33} Ce _{0.67}	1.8	450	6,000	100	372	49
Ru/CeO ₂	1.0	350	22,000	ca. 32.0	814	50
Ru/La ₂ O ₃	4.8	450	18,000	72.8	304.2	51
Ru-Mg(NH ₂) ₂	5.0	400	60,000	ca. 3~4	24.2	52
Ru-Ca(NH ₂) ₂	4.6	400	60,000	ca. 7~8	100	52
Ru-Ba(NH ₂) ₂	4.4	400	60,000	ca. 20	183.4	52
Ru/C12A7:e ^{-a}	2.2	400	15,000	70.0	532	26
Ru/C12A7:e ^{-b}	2.2	450	15,000	ca. 99.9	759	26
Ru/BHA ^c	2.74	450	60,000	ca. 20.8	507	42
Ru/MgO	2.8	450	30,000	41.3	493	53
Ru/MgO ^d	3.5	450	36,000	52.7	606	54
Ru/MgO-MIL ^e	3.1	450	15,000	ca. 70.0	377	17
Ru/c-MgO	4.7	450	30,000	80.6	565	32
Ru/Com-MgO	3.3	400	30,000	24.8	251.0	This work
Ru/MgO(100)	3.2	400	30,000	20.1	209.4	This work
Ru/MgO(110)	3.2	400	30,000	32.2	337.5	This work
Ru/MgO(111)	3.1	400	30,000	68.9	745.2	This work
Ru/MgO(111)	3.1	450	30,000	100.0	1080.6	This work

Ru/MgO(111)	3.1	450	60,000	82.2	1777.4	This work
Ru/MgO(111)	0.1	450	30,000	4.2	1400.0	This work

^a Reaction conditions: WHSV = 15,000 mL g⁻¹ h⁻¹, T = 400 °C; ^breaction conditions: GHSV = 15,000 mL g⁻¹ h⁻¹, T = 450 °C; ^cBHA: Barium hexaluminate, conditions: WHSV = 60,000 mL g⁻¹ h⁻¹; ^d reaction conditions: WHSV = 36,000 ml g⁻¹ h⁻¹, K/Ru=1/2; ^econditions: WHSV = 15,000 mL g⁻¹ h⁻¹, T = 450 °C.

5. Finally, I'm surprised by the claim of binuclear Ru pairs. I do not see them in Fig. 4. The catalysts labeled (4) and (5), which are the most active, contain a plethora of Ru ensembles. In fact, it is hard to recognize binuclear associations at all...

Reply: Thank you for the very constructive comment. By reanalyzing the data and interpretations, we are sorry for the misleading claim of binuclear Ru pairs in the original manuscript. From the modelling, we did not see any evidence of two NH₃ molecules simultaneously activate on the same Ru atom for N₂ formation. Instead, the modelling suggested that it would require at least two neighbor Ru atoms to generate the N₂ (see Fig 7). Also, from our experiments, we obtained a sharp surge in specific activity at higher Ru loading (TEM indeed indicated a plethora of surface Ru ensembles) beyond the isolated Ru species indicating that the synergetic effects of surface Ru atoms in this catalyst. However, it is not yet known whether the reaction requires the formation of two rigid defined Ru sites or surface clusters of Ru or even dynamic Ru sites on the MgO(111) surface in the Ru ensemble. As a result, we have changed the title from "Dispersed Binuclear Ru Pairs for Catalytic Ammonia Decomposition" to "Dispersed Surface Ru Ensembles on MgO(111) for Catalytic Ammonia Decomposition".

REVIEWER COMMENTS

Reviewer #1 (Remarks to the Author):

I would encourage the authors in the future to reproduce the revisions to the manuscript within the response letter, and further to provide a version of the manuscript with changes highlighted; both of these would simplify rereview. As written, the relationship between the response and the actual changes to the manuscript is very difficult to discern.

Below are my comments on the responses, numbered as my previous comments:

1. The narrative still states that Ru is the "best" for both ammonia synthesis and decomposition. Ref. 7 shows that "best" is a function of reaction conditions and is not necessarily the same for both reactions. The point of the manuscript is the support effect, so unclear that the comments regarding Ru position on a "volcano" are even necessary.

2. Table 1 still reports rates on a mass-normalized basis. Again, this metric, while relevant macroscopic performance and cost, does not provide information about intrinsic, site normalized rates and, without information about Ru dispersion, does not report clearly on support effects. Most of the discussion continues to revolve around these mass-normalized rates and comparisons of conversions; I can't easily find the TOF results in the manuscript. Table S7 reports that the support effect in the present work is about a factor of 4-5 on an exposed Ru basis, which I would not characterize as "much" higher. Do the dispersion values support the characterization of the material as atomically dispersed? Figure 4a makes arguments based on TOF's that differ by a factor of two, ie, very little. Are these results reproducible? What are the error bars on these rate measurements, considering all uncertainties? Are these comparisons truly robust?

3. Thanks for the addition

4. >15% conversion does not necessarily guarantee absence of equilibrium effects. I'd encourage the authors to avoid hyperbolic words, for instance "remarkable" in comparing activation energies.

5. The scheme needs to be incorporated into the narrative itself to be helpful. As written, the narrative contains substantial speculation regarding this mechanistic behavior (which should be properly recognized as such) and is very difficult to follow without a scheme.

6. Thanks for the addition

7. The presentation of the DFT calculations is improved; thanks to the authors for the effort here. The heart of the manuscript is the idea that H "spills over" to MgO(111), so that H does not "poison" Ru. For this to be correct, that H would have to leave the support as H₂. Was that process computed using DFT? Manuscript alludes to H migrating back to Ru to recombine. If this was the case, then spillover would not be kinetically relevant. Some further explanation is necessary. Can a microkinetic model based on the DFT rationalize the observed relative kinetics on different supports.

Overall, this would be a much stronger manuscript if the authors focused more on careful, measured comparisons and less on rationalizations of results that have a questionable basis.

Reviewer #2 (Remarks to the Author):

The authors have done a good job in answering to all the issues raised by the referees. I'm particularly pleased with the answer regarding the amount of the different facets exposed by MgO; the authors have added valuable data (DRIFTS with CO₂ as a probe molecule) to measure their relative amount. They have also added atomic-resolved EELS to unambiguously identify the occurrence of Ru atoms. In my opinion the work can now be considered for publication. I just have a couple of additional comments:

1. Table 1, which has now been expanded, should be moved to the Supplementary Information as it is a long table intended to compare the catalytic performance of catalysts other than those reported in this work.
2. For the sake of clarity of the catalytic results reported in the literature, the authors may consider adding in the introduction of this work, a review recently appeared in the literature, which contains abundant and valuable data: <https://pubs.acs.org/doi/10.1021/acs.iecr.1c00843>

Reviewer #1 (Remarks to the Author):

I would encourage the authors in the future to reproduce the revisions to the manuscript within the response letter, and further to provide a version of the manuscript with changes highlighted; both of these would simplify rereview. As written, the relationship between the response and the actual changes to the manuscript is very difficult to discern.

We sincerely thank this Reviewer for evaluating our work again. We apologize that we did not do the reproduction of the revisions in the reply and caused difficulties to him/her for discerning our changes in the manuscript. We have now prepared the response letter and revised manuscript with highlights for your further review.

Below are my comments on the responses, numbered as my previous comments:

1. The narrative still states that Ru is the "best" for both ammonia synthesis and decomposition. Ref. 7 shows that "best" is a function of reaction conditions and is not necessarily the same for both reactions. The point of the manuscript is the support effect, so unclear that the comments regarding Ru position on a "volcano" are even necessary.

Reply: Thank you for raising the good comment. We understand the concerns from this Reviewer (also from his/her question 7) whether the decomposition reaction is classified as 'structural sensitive' or 'insensitive', Ru is the best metal for volcano plots under standard conditions since in the case of synthesis, N₂ cleavage and in the case of decomposition, N-N recombination, as RDS which depends on the Ru-N bonding characteristics. In principle, it should be structural insensitive (ignoring the structural sensitivity of Ru metal sites) but recent studies for reaction kinetics particularly under milder conditions adaptive to renewables, are found to depend on reaction conditions (Ref 7) and from us, the type of support used hence can be 'structural sensitive' indicative that the other kinetics such as H migration may become rate limiting on some reaction conditions with the particular type of supports used. This issue is somehow similar to the recent kinetic study of Fischer–Tropsch reaction (*ACS Catal.* 2019, 9, 5, 4189–4195). While the dissociation of the C–O bond is well-known to be crucial over metal in determining the overall reaction rate, recent experimental results also show that a hydrogenation step is also crucial in the overall kinetics particularly at lower H₂ pressure.

In our cases, the "best" for catalytic performance under milder conditions is indeed strongly connected to the reaction conditions particularly at lower temperature and pressure. Under the same conditions, support effect to remove H poisoning is crucially important. To avoid misunderstanding that the kinetics must link with the volcano optimum at Ru disregarding other effects, we changed our description in the manuscript as follows:

Action taken:

The original description "However, ruthenium (Ru) at the optimal position of

volcano activity over transition metal of increasing d orbital electrons, has been well-recognized as the most efficient metal for both reactions with many folds higher activity than other transition metals.” *has been changed to* **“Ruthenium (Ru) has been well-studied as an efficient metal for ammonia decomposition at low temperatures due to its catalytic performance with many folds higher activity than other transition metals.”** (Page 2)

2. (A) Table 1 still reports rates on a mass-normalized basis. Again, this metric, while relevant macroscopic performance and cost, does not provide information about intrinsic, site normalized rates and, without information about Ru dispersion, does not report clearly on support effects. Most of the discussion continues to revolve around these mass-normalized rates and comparisons of conversions; I can't easily find the TOF results in the manuscript. (B) Table S7 reports that the support effect in the present work is about a factor of 4-5 on an exposed Ru basis, which I would not characterize as "much" higher. (C) Do the dispersion values support the characterization of the material as atomically dispersed? (D) Figure 4a makes arguments based on TOF's that differ by a factor of two, ie, very little. Are these results reproducible? What are the error bars on these rate measurements, considering all uncertainties? Are these comparisons truly robust?

Reply: Thanks for the reviewer's kind suggestion and comment.

(A) We agree with the reviewer. According to the Reviewer's suggestion, we have tried our best to work out the TOF values from literature data and ours in revised Table S1 for better comparison of the intrinsic activities over different catalysts. The detailed Ru dispersion values in this work were demonstrated in Table S9. We have also revised our discussion based on the TOF values. Please see the highlights in the revised manuscript.

(B) Thanks for the kind suggestion. We have carefully checked our description and deleted hyperbolic expressions for proper presenting our work.

(C) We first carried out XAS analysis and fitting (Fig. S15 and Table S2) to ensure the atomic dispersion of Ru on MgO as those shown by the discrete atom dispersions in the STEM images. The average coordination number of surface Ru ensembles is 2.6 in the Ru/MgO(111) sample. These results also confirmed that the catalyst surface actually contains a plethora of total dispersed Ru ensembles (no formation of clusters or nanoparticles). We have also performed the CO chemisorption to determine the Ru dispersion for the calculation of TOF values. Although the calculated Ru dispersion based on CO chemisorption are lower than 100%, which is presumably due to some Ru atoms could be covered by the support, the value indicated the approaching to atomic dispersion. Previous reports (e.g., Nano Lett. 2018, 18, 3785–3791; Nat. Commun. 2022, 13, 3188) also reported similar results that the metal dispersion via chemisorption in their single-atom catalysts are still lower than 100%. Notably, our TOF values for comparison are calculated based on the actual Ru dispersion values via CO chemisorption.

(D) We understand the concerns from this Reviewer. The results shown in Fig. 4a are reproducible within experimental errors based on at least three repeated measurements.

The repeated results are similar and only fluctuated in a narrow range of conversion, which did not affect the average TOFs. In light of his/her comment, we have added the error bars in Fig.4a to ensure that the comparison is reliable. The Fig. 4 has been revised in the manuscript.

Action taken:

(1) The **Table 1** *has been revised* as follows.

Table S1 NH₃ conversion over Ru catalysts under atmospheric pressure.

Catalyst	Ru / wt%	T / °C	WHSV / mL g _{cat} ⁻¹ h ⁻¹	Conv. / %	H ₂ formation rate / mmol g _{Ru} ⁻¹ min ⁻¹	TOF _{NH₃} ^e / s ⁻¹	Ref.
Ru/Al ₂ O ₃	10	450	30,000	31.5	115	4.6	21
Ru/SiO ₂	10	450	30,000	34.5	114	11.3 ^f	21
Ru/MCM-41	5	450	30,000	42.4	284	2.3	22
Ru@ZrO ₂	3	450	30,000	ca. 40	447	–	23
Ru/CNTs	5	450	30,000	43.7	292	0.75	24
Ru/MgO	4.8	450	30,000	30.8	215	2.6	25
Ru/TiO ₂	4.8	450	30,000	27.2	190	2.1	25
Ru/Al ₂ O ₃	4.8	450	30,000	23.3	163	1.6	25
Ru/AC	4.8	450	30,000	28.7	200	1.7	25
Ru/CNFs	3.2	500	6,500	99.0	206	0.23	26
Ru/Cr ₂ O ₃	5.0	600	30,000	99.9	667.5	0.12	27
Ru/La _{0.33} Ce _{0.67}	1.8	450	6,000	100	372	1.8	28
Ru/CeO ₂	1.0	350	22,000	ca. 32.0	814	2.13	29
Ru/La ₂ O ₃	4.8	450	18,000	72.8	304.2	1.5	30
Ru-Mg(NH ₂) ₂	5.0	400	60,000	ca. 3~4	24.2	0.09	31
Ru-Ca(NH ₂) ₂	4.6	400	60,000	ca. 7~8	100	0.28	31
Ru-Ba(NH ₂) ₂	4.4	400	60,000	ca. 20	183.4	0.86	31
Ru/C12A7:e ^{-a}	2.2	400	15,000	70.0	532	6.9	32
Ru/C12A7:e ^{-b}	2.2	450	15,000	ca. 99.9	759	6.9	32
Ru/BHA ^c	2.74	450	60,000	ca. 20.8	507	2.9	33
Ru/MgO	2.8	450	30,000	41.3	493	0.84	34
Ru/MgO ^d	3.5	450	36,000	52.7	606	2.0	35
Ru/MgO-MIL ^e	3.1	450	15,000	ca. 70.0	377	–	36
Ru/c-MgO	4.7	450	30,000	80.6	565	2.6	37
Ru/Com-MgO	3.3	400	30,000	24.8	251.0	0.71	This work
Ru/MgO(100)	3.2	400	30,000	20.1	209.4	0.59	This work
Ru/MgO(110)	3.2	400	30,000	32.2	337.5	0.90	This work
Ru/MgO(111)	0.08	400	30,000	4.5	1902.0	3.43	This work
Ru/MgO(111)	0.94	400	30,000	31.2	1109.9	2.25	This work
Ru/MgO(111)	3.1	400	30,000	68.9	745.2	2.33	This work

Ru/MgO(111)	3.1	450	30,000	100.0	1080.6	4.91	This work
Ru/MgO(111)	3.1	450	60,000	82.2	1777.4	4.91	This work

^a Reaction conditions: WHSV = 15,000 mL g⁻¹ h⁻¹, T = 400 °C; ^breaction conditions: GHSV = 15,000 mL g⁻¹ h⁻¹, T = 450 °C; ^cBHA: Barium hexaluminate, conditions: WHSV = 60,000 mL g⁻¹ h⁻¹; ^d reaction conditions: WHSV = 36,000 ml g⁻¹ h⁻¹, K/Ru=1/2; ^econditions: WHSV = 15,000 mL g⁻¹ h⁻¹, T = 450 °C; ^e obtained from Ref., estimated based on the published data or calculated by experimental data; ^f the metal dispersion is 1.2%.

(2) Detailed discussions on TOF values were incorporated in the revised manuscript.

It is presented that the atomically dispersed surface Ru ensembles on MgO(111) surface gives an optimal turnover frequency (TOF) value of 3.43 s⁻¹ for ammonia decomposition at 400 °C, higher than those of allegorically more active Ru cluster/nanoparticle counterparts on MgO(111) (2.1–2.9 s⁻¹) in classic view and the ultralow-loading Ru/MgO(111) with isolated Ru atoms (1.67 s⁻¹). (Page 3)

It is worth noting that the TOF value of Ru/MgO(111) is 2.33 s⁻¹ at 400 °C, higher than that of Ru/MgO(110) (0.90 s⁻¹) and Ru/MgO(100) (0.59 s⁻¹), manifesting the importance of exposed MgO facets on catalytic performance. (Page 9-10)

At low Ru loading, the TOF value rapidly increases with increase of the surface Ru loading, achieving the maximum TOF value of 3.43 s⁻¹ and 4.91 s⁻¹ at 400 and 450 °C, respectively. (Page 18)

(3) We have deleted the hyperbolic expressions such as “much”, “remarkably”, “superior” and so on.

(4) The **Fig. 4a** *has been* revised in the manuscript.

Fig. 4 (a) The TOF values for ammonia decomposition as a function of Ru loading at 400 °C; (b) Typical STEM images of Ru/MgO(111) with different loadings, the labeled numbers are corresponding to the samples in Fig. 4(a); the scale bar is 1 nm.

3. Thanks for the addition

Reply: Thank the reviewer for his/her satisfaction with our revision.

4. >15% conversion does not necessarily guarantee absence of equilibrium effects. I'd encourage the authors to avoid hyperbolic words, for instance "remarkable" in comparing activation energies.

Reply: Thanks for the reviewer's kind suggestion and agreed with his reminder. We have modified the statement and deleted the hyperbolic expressions and revised our description in the revised manuscript.

Action taken:

We have deleted the hyperbolic expressions such as "highest", "much", "remarkably", "superior" and so on.

5. The scheme needs to be incorporated into the narrative itself to be helpful. As written, the narrative contains substantial speculation regarding this mechanistic behavior (which should be properly recognized as such) and is very difficult to follow without a scheme.

Reply: Thank the reviewer for his/her nice suggestion. We have moved the Scheme 1

into the manuscript and added more discussion for better understanding for readers.

Action taken:

The sentences and the Scheme 1 have been added in the revised manuscript.

Scheme 1 The schematics demonstrating the difference between the (a) Ru/MgO(111) and (b) Ru/MgO(100).

Scheme 1 is depicted to further illustrate the varied reaction order between different MgO supported Ru catalysts. Compared with MgO(100), the exposed MgO(111) surface exhibits a strong affinity for hydrogen activation (via Frustrated Lewis pair with Ru species) and removal (proton hopping), corresponding to its smaller γ value, which helps to reduce the hydrogen poisoning. In addition, the reduction of Ru during hydrogen activation can also promote the N–N recombination by donating the electrons to the Ru–N antibonding orbitals, corresponding to the large α value.

6. Thanks for the addition

Reply: Thank the reviewer for his/her satisfaction with our revision.

7. The presentation of the DFT calculations is improved; thanks to the authors for the effort here. The heart of the manuscript is the idea that H "spills over" to MgO(111), so that H does not "poison" Ru. For this to be correct, that H would have to leave the support as H₂. Was that process computed using DFT? Manuscript alludes to H migrating back to Ru to recombine. If this was the case, then spillover would not be kinetically relevant. Some further explanation is necessary. Can a microkinetic model based on the DFT rationalize the observed relative kinetics on different supports. Overall, this would be a much stronger manuscript if the authors focused more on careful, measured comparisons and less on rationalizations of results that have a questionable basis.

Reply: We understand the concerns from this Reviewer, we think that this question is somehow related to his/her question 1. As stated, most studies on the ammonia decomposition reaction in literature assume that the recombination of the N–N bond is crucial in determining the overall reaction rate when the reaction temperature and hydrogen pressure are high where the hydrogen orders (γ) of reaction are positive.

However, experimental results show that the hydrogen migration step (avoid poisoning) is crucial in the overall kinetics when much milder conditions are used. As we have showed that at low pressures, which is more relevant for renewable energy transport, the support interaction becomes important. For example, in Fig. 2e, the hydrogen orders (γ) for both Ru/MgO(110) and Ru/MgO(100) were large and negative, reaching -0.79 and -0.85 , respectively. This indicates that hydrogen adatoms are strongly adsorbed onto the Ru surface, well-known as hydrogen poisoning, preventing further activation and chemisorption of ammonia and nitrogen species on these competitive Ru sites. (structure-sensitive). On the other hand, the use of polar MgO(111) surface could render the kinetics much less dependent on hydrogen (spillover readily achieved) under the milder conditions. Under these conditions, the obtained activation energy and the negligible N_2 pressure dependence on reaction rate suggest that the N recombination is still RDS, in agreement with the previous conclusion for other catalysts under high temperature/pressure conditions. Although we have not computed the H, H recombination to H_2 due to the fact that the rate limiting is in the N-N recombination under this catalyst and conditions used, we agree that developing a microkinetic model based on DFT could be very useful to rationalize the observed kinetics on different supports and particularly under different reaction conditions. We have added the above comment in the revised manuscript to indicate our future study.

Action taken:

The discussion *has been added* in the revised manuscript as follows.

Our experimental results show that the hydrogen migration step (avoid poisoning) is crucial in the overall kinetics when much milder conditions are used. As we have showed that at low pressures, which is more relevant for renewable energy transport, the support interaction becomes important. The use of polar MgO(111) surface could render the kinetics much less dependent on hydrogen (spillover readily achieved) under the milder conditions; on the other hand, hydrogen adatoms are strongly adsorbed onto the Ru surface, well-known as hydrogen poisoning, was further observed on both Ru/MgO(110) and Ru/MgO(100) (Fig. 2e), preventing further activation and chemisorption of ammonia and nitrogen species on these competitive Ru sites. It is believed that spillover H will likely to go back to Ru sites to acquire electrons and produce H_2 but this process is fast and does not affect the experimentally observed ammonia decomposition activity. Thus, to avoid distraction, we have excluded the H_2 desorption step in the DFT simulations although we believe that developing a microkinetic model based on DFT could be very useful to rationalize the observed kinetics on different supports and particularly under different reaction conditions. Instead, we have chosen to account for the effect of the unremoved of H^* atoms on Ru for $Ru_2/MgO(100)$ and $Ru_2/MgO(110)$ and spillover H-O on $Ru_2/MgO(111)$ on the thermodynamics of the steps toward N-N recombination. (Page 24)

Reviewer #2 (Remarks to the Author):

The authors have done a good job in answering to all the issues raised by the referees. I'm particularly pleased with the answer regarding the amount of the different facets exposed by MgO; the authors have added valuable data (DRIFTS with CO₂ as a probe molecule) to measure their relative amount. They have also added atomic-resolved EELS to unambiguously identify the occurrence of Ru atoms. In my opinion the work can now be considered for publication. I just have a couple of additional comments:

We sincerely thank the reviewer for evaluating our manuscript again and improving our work with constructive suggestions.

1. Table 1, which has now been expanded, should be moved to the Supplementary Information as it is a long table intended to compare the catalytic performance of catalysts other than those reported in this work.

Reply: Thanks for the reviewer's kind suggestion. We have moved the Table 1 into the Supplementary Information.

Action taken:

The Table 1 *has been modified and moved* to the Supplementary Information.

2. For the sake of clarity of the catalytic results reported in the literature, the authors may consider adding in the introduction of this work, a review recently appeared in the literature, which contains abundant and valuable data:

<https://pubs.acs.org/doi/10.1021/acs.iecr.1c00843>

Reply: Thanks for the reviewer's kind suggestion. We have added the reference and related discussion in the revised manuscript.

Action taken:

(1) The Reference "**Lucentini, I., Garcia, X., Vendrell, X., and Llorca, J. Review of the decomposition of ammonia to generate hydrogen. *Ind. Eng. Chem. Res.* 60, 18560–18611 (2021).**" *has been added* in the revised manuscript.

(2) The sentence "**Ilaria systematically summarized the recent progress in developing Ru based catalysts for ammonia decomposition and these Ru catalysts demonstrated a variety of catalytic performance due to their different supports, metal structure and particle size.**" *has been added* in the revised manuscript.

According to the Reply, we also reorganized the order of Figures, Tables, Scheme and References.

REVIEWER COMMENTS

Reviewer #3 (Remarks to the Author):

The authors have well addressed the comments of previous referees and have made improvements to the quality of the manuscript. But I have several questions on the DFT studies.

1. In Fig. 5a, during the combination of two N^* to form N_2^* , the lattice oxygen was pulled out by the Ru and becomes $Ru=O^*$ species, as can be easily found in states I, II, and III. Consequently, restoring the initial surface structure from the state III requires an energy cost of almost 6 eV, which means that it is thermodynamically almost impossible to happen under the reaction conditions. Please explain. If the adsorption of NH_3 may help to restore the surface, the authors should add related discussions on this point. And if this is the case, the state III, as the most stable surface structure, should be used as the initial state for the catalytic cycle.

2. Did the authors consider the possible structure of Ru embedded in the surface lattice of MgO (replacing Mg) rather than supported on the surface?

3. In Fig. 5b, it is hard to obtain useful information from the messy DOS plotted with so many lines. The authors may need to make clearer and neater DOS figures to help understanding their study, i.e. the quality of the DOS figures need to be largely improved.

Reviewer #3 (Remarks to the Author):

We appreciate the comments from the reviewer. We have revised the discussion of the DFT data according to your suggestions. Please find our point-by-point response below. We hope that these changes can sufficiently address your concerns.

1. In Fig. 5a, during the combination of two N^* to form N_2^* , (1) the lattice oxygen was pulled out by the Ru and becomes $Ru=O^*$ species, as can be easily found in states I, II, and III. (2) Consequently, restoring the initial surface structure from the state III requires an energy cost of almost 6 eV, which means that it is thermodynamically almost impossible to happen under the reaction conditions. (3) Please explain. If the adsorption of NH_3 may help to restore the surface, the authors should add related discussions on this point. And if this is the case, the state III, as the most stable surface structure, should be used as the initial state for the catalytic cycle.

Reply: We thank the reviewer for this comment.

(1) Sorry for the misleading results presented in the original Fig. 5 as we did not properly explain such results and also the further modification (original Fig. 7). Initially, we calculated the N-N recombination steps on the $2Ru/MgO(111)$ (two neighboring single Ru atoms dispersed on $MgO(111)$ support) without H adatoms since we would like to focus on the mechanism of N_2 formation step, which led to the reconstruction (the lattice oxygen was pulled out by the Ru and becomes $Ru=O^*$ species) of the $MgO(111)$ surface. We totally understand the Reviewer's concern. Actually, oxygen spillover phenomenon (oxygen pulled out) is often observed in systems with strong metal-support interaction (SMSI) in previous studies (e.g., *J. Chem. Phys.* 2019, 151, 204703; *Surf. Sci.* 2016, 646, 230–238).

We had also considered a more reasonable situation as shown in revised Fig. 5a (Fig. 7 in the previous revision). The models of $2Ru/MgO(111)$ system with six hydrogen adatoms on the support exhibited significant decreased $Ru=O^*$ formation due to the favorable formation of surface $-OH^*$ species, which is in agreement with our DRIFT and AP-XPS results. In the new DFT results, the O spillover phenomenon has significantly reduced after loading six hydrogen atoms in the models; we cannot avoid the O spillover completely in this model since not all O atoms are covered by hydrogen atoms. However, in real experimental result, the abundant hydrogen atoms from ammonia decomposition is expected to spillover to the support extensively and inhibit the reconstruction (oxygen protrusion) of $MgO(111)$ and the formation of surface $Ru=O^*$ species under conditions. **To avoid misunderstanding for reviewers and readers, the Fig. 5 has now been renewed in the revised manuscript as follows. We have also reorganized our DFT discussion for clearer description and better understanding for readers on our modeling analysis. See revised manuscript with highlights for details.**

Fig. 5 DFT calculated energy profiles and their corresponding optimized intermediate structures for (a) ammonia decomposition and N–N recombination pathways to form N_2 at MgO-supported two Ru single atoms and (b) hydrogen removal from NH^* species and hydrogen spillover at MgO-supported Ru single atom. The reactions were calculated over (100), (110) and (111) facet of MgO shown in blue, green and red lines, respectively. E_a (in eV) refers to the activation energies of hydrogen spillover from the NH^* species to the three MgO supports. The optimized structure of each intermediate is depicted wherein Mg, O, Ru, N and H atoms are depicted as orange, red, purple, blue and gray spheres, respectively; the values near the atoms of H, N and Ru in the inserted structure are their respective Bader charges (in |e|). The value near each intermediate refers to their corresponding adsorption energy (in eV), with respect to a single H_2 and N_2 gas.

In addition, we modified the original Fig. 5 by calculating the barrier of N-N recombination in the presence of spillover H adatoms and moved it in Fig. S18 in the supporting information. The revised Fig. S18 still confirms that the rate-determining step is the N-N recombination with a barrier of 1.49 eV. Thus, the conclusions regarding the kinetics of ammonia decomposition does not change with such revised models.

Fig. S18 DFT calculated energy profile of the ammonia decomposition pathway on Ru pair sites supported on MgO(111). The N-N recombination step with a calculated activation energy value of 1.49 eV (TS5) is the rate-determining step.

(2) Thank you for the reviewer pointing it out. We also understand the Reviewer's concern for high energy cost to restore the initial surface structure from the state III in the original manuscript. For a SMSI system, the configuration with oxygen spillover is usually more stable than non-oxygen spillover configuration. If the O-terminated MgO(111) surface could ideally exist without hydrogenation, the large energy difference between N₂-2Ru/MgO(111) and N₂ + 2Ru/MgO(111) configurations reveal the following facts: If we do not consider the scenario of atomic H on MgO(111), (a) during the catalytic reaction, the morphology of catalysts would possibly be altered. (b) oxygen spillover onto the Ru catalyst surface will stabilize the entire catalysts for a SMSI catalyst. Thus, in the last step of N₂ detachment, in the original Fig. 5a, probably we shall not compare the configurations between O spillover and non-O spillover surface, as the reviewer mentioned, it is thermodynamically almost impossible to recover the original catalyst.

Instead, we have modified our original results to include more reasonable models with six H adatoms from two NH₃ molecules as shown in revised Figs. 5a and S18. In the last step, it is obvious to see that the breakage of the last Ru-N bond to form N₂ is much more thermodynamically favorable in MgO(111), compared to the other non-polar (100) and (110) facets in Fig 5a. Also, we can see the reaction energy of the last step from N₂-2Ru/MgO(111) to N₂ + 2Ru/MgO(111) reduced to 1.37 eV (with six H atoms on MgO(111)), compared to 6 eV (without six H atoms on MgO(111)), indicating the fact that saturation of hydrogen of MgO(111) will inhibit the occurrence of O spillover that might not really exist in reality.

(3) As explained above, the O spillover (protrusion) can be inhibited by the presence of spillover H adatoms onto MgO(111) surface. In the revised Fig 5a, less O spillover (the lattice oxygen was pulled out by the Ru) is seen. However, in such DFT models, we cannot avoid the O spillover completely since not all O atoms are covered by hydrogen atoms. Nevertheless, we believe that, in real experimental conditions, H adatoms might saturate the MgO(111) to stabilize the catalyst surface, corresponding to our previous experimental observation (*ACS Catal.* 2020, 10 (10), 5614–5622; *J. Am. Chem. Soc.* 2021, 143 (24), 9105–9112). We have reorganized our discussion in the revised manuscript for better understanding.

2. Did the authors consider the possible structure of Ru embedded in the surface lattice of MgO (replacing Mg) rather than supported on the surface?

Reply: We thank the reviewer for raising this question.

Accordingly, we consider the possibility of embedded Ru structure into two situations, one is that a periodic crystal has been formed; another is that there exists non-periodic segregation in the crystal. Obviously, the situation of periodic infiltration of Ru into the MgO crystal is insufficient, because the x-ray diffraction does not change significantly compared with the pure MgO powder. If the Ru has replaced Mg at a long-range order, new diffractogram can be recognized by x-ray diffraction. In our case, X-ray absorption fine structure (EXAFS) shown Fig. 1g and subsequent least square fitting reveals that Ru is coordinated with 3 oxygen atoms. For non-periodic segregation in the crystal, the crystallographic phases of Ru embedded in the surface lattice of MgO can be verified through higher spatial resolution of TEM image. We captured signals of Ru atom sites from the surface in different crystallographic orientations of MgO support in Fig. S5. The heavy Ru atoms as bright dots surrounded by 3 O atoms on the surface of MgO(111) and the line mapping images confirm that Ru atoms are found on the OOO hollow site on the MgO(111) surface. In addition, the contrast intensity of Ru atom in a column containing the same number of Mg atoms will obviously exceed the contribution of the surrounding Mg. The significant contrast enhancement was not found at the edges of the MgO particles either. Thus, all these observations are consistent with the binding of Ru on the MgO(111) surface but not the embedded Ru model.

In addition, as suggested by the reviewer, we have performed DFT calculations to compare the stability of Ru adsorbed on the MgO(111) surface compared with Ru embedded in place of a subsurface Mg in the MgO(111) slab model. These new results, presented in the revised Fig. S16a, show that the formation energy is more negative for Ru adsorbed on MgO(111) than that for Ru embedded into MgO(111). This confirms that Ru adsorbed on MgO(111) model is more thermodynamically favorable than the embedded Ru model. We have also improved the description of our computational details in the supplementary information.

Fig. S16a Optimized structure and formation energies (E_f , in eV) of a single Ru atom (A) adsorbed on MgO(111) (O-O-O hollow) site and (B) embedded in place of an Mg atom of MgO(111).

3. In Fig. 5b, it is hard to obtain useful information from the messy DOS plotted with so many lines. The authors may need to make clearer and neater DOS figures to help understanding their study, i.e., the quality of the DOS figures need to be largely improved.

Reply: We are grateful for the reviewer for pointing this out. We agree that it is difficult to abstract meaningful interpretations from the messy nature of the projected DOS (PDOS) of different Ru d orbitals in the original manuscript. We have recalculated and redrawn the PDOS (we also replace the terminology of LDOS by PDOS) by using the models without O spillover according to the comment and present it as Fig. 6 in the revised version. **The revised Fig. 6 shows newly generated PDOS and partial charge plots of the corrected models of the N-N recombination intermediates on 2Ru/MgO(111). We have also improved the readability of the DOS plots by using a color scheme corresponding with the images of**

the optimized intermediate structures. We believe the revised Fig. 6 much more effectively shows the decoupling of the Ru and N states as N₂ is formed. The revised Fig. 6 is copied below.

Fig. 6 Optimized structures, partial density of state and partial charge density starting from 2 surface Ru≡N to N₂ via μ-η¹:η² bimetallic nitrido-species complex like structure in homogeneous systems^{28,62,64} as intermediate for the ammonia decomposition over the Ru/MgO(111).

We have also revised the discussions of the DFT data in the main text according to the revised figures. We note that the overall conclusions of the DFT section and its strong agreement with the experimental data remains the same. Please refer to the revised manuscript to see the minor changes made to the discussions of the DFT results under the “DFT calculations” section. We hope these revisions sufficiently address the concerns of the reviewer regarding our models.

A separate point-by-point response to the reviewers' comments, reproduced verbatim.

REVIEWERS' COMMENTS

Reviewer #3 (Remarks to the Author):

The authors have well addressed my concerns. I recommend publication of the manuscript in the journal.

Authors: we thank for the support from this Reviewer who has confirmed acceptance of this paper.